# What a MESS: Multi-Domain Evaluation of Zero-Shot Semantic Segmentation

**Benedikt Blumenstiel**[*]
Karlsruhe Institute of Technology
IBM Research Europe
benedikt.blumenstiel@kit.edu

**Johannes Jakubik**[*]
Karlsruhe Institute of Technology
IBM Research Europe
johannes.jakubik@kit.edu

**Hilde Kühne**
University of Bonn
MIT-IBM Watson AI Lab
hildegard.kuehne@ibm.com

**Michael Vössing**
Karlsruhe Institute of Technology
IBM Germany
michael.voessing@kit.edu

## Abstract

While semantic segmentation has seen tremendous improvements in the past, there are still significant labeling efforts necessary and the problem of limited generalization to classes that have not been present during training. To address this problem, zero-shot semantic segmentation makes use of large self-supervised vision-language models, allowing zero-shot transfer to unseen classes. In this work, we build a benchmark for Multi-domain Evaluation of Semantic Segmentation (MESS), which allows a holistic analysis of performance across a wide range of domain-specific datasets such as medicine, engineering, earth monitoring, biology, and agriculture. To do this, we reviewed 120 datasets, developed a taxonomy, and classified the datasets according to the developed taxonomy. We select a representative subset consisting of 22 datasets and propose it as the MESS benchmark. We evaluate eight recently published models on the proposed MESS benchmark and analyze characteristics for the performance of zero-shot transfer models. The toolkit is available at https://github.com/blumenstiel/MESS.

## 1 Introduction

Zero-shot semantic segmentation utilizes self-supervised models such as CLIP to minimize labeling requirements during training and to improve model generalization. Recent models are already able to include classes during inference that were not present during training. For this reason, zero-shot semantic segmentation is becoming increasingly relevant for real-world scenarios. In particular, the performance on domain-specific datasets such as earth monitoring datasets, as visualized in Figure 1, becomes more and more relevant. Current standard benchmarks tend to focus on in-domain tasks but do not capture performance comparisons across domains. This is problematic because it limits insight into the applicability of zero-shot semantic segmentation to new domains. It also makes it difficult to assess whether architectures might be suitable for datasets that pose additional challenges (e.g., different sensor types or specialized vocabulary). To better understand the behavior of zero-shot semantic segmentation models on a wider range of more complex, domain-specific datasets, we propose a holistic Multi-domain Evaluation of Semantic Segmentation (MESS). To this end, we have examined 120 datasets and classified them within a developed taxonomy. We leverage our benchmark to evaluate eight recently published models for zero-shot semantic segmentation including

---

[*]Equal contributions

37th Conference on Neural Information Processing Systems (NeurIPS 2023) Track on Datasets and Benchmarks.

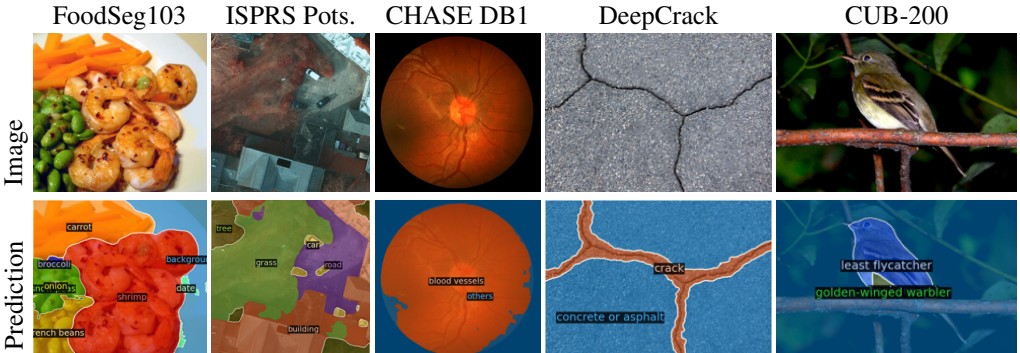

Figure 1: CAT-Seg-L [7] predictions for a range of domain-specific datasets. The model achieves promising predictions on everyday and satellite images, while it faces difficulties in segmenting small segments such as blood vessels, and distinguishing similar classes such as bird species.

the state-of-the-art models on 22 datasets from the fields of medical sciences, earth monitoring, agriculture and biology, engineering as well as a general domain including datasets on, e.g., driving scenes, maritime scenes, paintings, and body parts. Our evaluation focuses on zero-shot text-to-mask models—also known as open-vocabulary semantic segmentation (OVSS)—and later also compares their performance with zero-shot point-to-mask and box-to-mask approaches of SAM [20]. Using the proposed benchmark, we identify and analyze several characteristics that influence the performance of OVSS models, i.a., showing that the semantic, textual similarity of classes as well as the underlying sensor type, significantly affect the performance of current models.

Our experiments reveal various challenges for the application of zero-shot semantic segmentation on domain-specific datasets, e.g., we found that the selection of class labels can significantly affect the quality of predictions. We also observe that the models are sensitive to the semantics of the textual prompts, e.g., general terminology leads to better performance than domain-specific terminology. Overall, we hope that our benchmark will support accelerating zero-shot semantic segmentation and improve the real-world applicability of semantic segmentation in general.

We summarize the contributions of this work as follows: (1) We develop a taxonomy based on a quantitative and qualitative analysis of a broad variety of semantic segmentation datasets. (2) We propose a new benchmark for multi-domain semantic segmentation. (3) We evaluate eight zero-shot models on the MESS benchmark with an in-depth analysis of the task characteristics.

## 2 Related work

### 2.1 Zero-shot semantic segmentation

Large-scale self-supervised pre-training has revolutionized the field of computer vision over the last couple of years. One stream of work focuses on vision-language pre-training such as in recent foundation model architectures like CLIP [37], ALIGN [19], and Florence [54]. These models are trained on image-text pairs and encode both visual and text semantics in a shared embedding space. This approach particularly enables so-called open-vocabulary image classification by computing the similarity between the embeddings of the image and the embeddings of natural language describing the classes in the image. The text describing the images can be any arbitrary textual sequence and might describe classes on the images that have been unseen during training. This is in contrast to recent segmentation models, like Segment-Anything (SAM, see [20]), which are trained only on image data and therefore do not include a text encoder to encode semantic concepts. Hence, segmentation models like SAM do not facilitate open-vocabulary out of the box and need to be adapted to support the processing of textual information (e.g., by using additional models that generate text embeddings or models that provide bounding boxes as input such as Grounding DINO [27]).

Early approaches in OVSS have been built upon standard zero-shot semantic segmentation, such as ZS3Net [5], using simple word2vec text encoders. Subsequent two-stage approaches made use of mask proposals based on MaskFormer [6] in stage 1 followed by predictions of each mask by

CLIP [9, 10, 25, 51]. Recently, one-stage frameworks like SAN [50] generate masks in a side adapter network during the CLIP inference. Therefore, CLIP does not classify many mask proposals but only the image ones, resulting in a faster inference. Other mask-based models like GroupViT [49] and ViewCo [40] are grouping pixels into larger segments which are then classified.

Decoder-focused approaches such as DenseCLIP [39] and LSeg [23] encode the image with CLIP and obtain the pixel-level patch embeddings. Because the pre-training is focused on the class embedding, the approaches append additional decoders to refine the patch embeddings. For this refinement, CAT-Seg [7] utilizes multiple stages of cost aggregation to generate the final segmentation mask. PACL [34] aligns patch embeddings and class embeddings during training and, as a result, does not require segmentation-specific training data or additional modules. Zero-shot semantic segmentation models have been combined with other tasks as well. OpenSeeD [55] implements open-vocabulary for object detection and segmentation. SEEM [58] processes text prompts and additional inputs like visual prompts similar to SAM. Apart from differences in the architecture, the models vary in the training process—particularly in fine-tuning CLIP's vision encoder.

## 2.2 Evaluation and benchmarking of zero-shot semantic segmentation

Zero-shot semantic segmentation models are typically evaluated on datasets consisting exclusively of everyday images, such as ADE20K [56], Pascal Context [33], and Pascal VOC [12]. These dataset are the *de facto* standard for evaluating these models (see [7, 14, 25, 50, 51, 57]). Few studies have considered additional datasets. Notably, Zou et al. [57] proposed a Segmentation in the Wild (SegInW) benchmark with 25 datasets. However, the majority of the datasets in SegInW still consist of everyday images with only two exceptions: brain tumor segmentation and a bird's eye view in stables. To the best of our knowledge, zero-shot semantic segmentation and OVSS have not been evaluated on other datasets. Outside of zero-shot semantic segmentation and OVSS, semantic segmentation is usually evaluated based on collections of datasets, like MSeg [22]. These datasets generally only include everyday images, indoor scenes, and driving datasets and lack domain-specific datasets. SAM has been evaluated on 23 instance segmentation datasets in a point-to-mask setting [20]. This collection of datasets is the most extensive for segmentation tasks but still misses domains, such as engineering and earth monitoring. Other works evaluate specifically domain dataset collections such as medical tasks [32] or satellite data [21].

Table 1: Multi-domain benchmark for zero-shot semantic segmentation models consisting of 22 downstream tasks, a total of 448 classes, and 25,079 images.

| Dataset | Domain | Sensor type | Segment size | Number of classes | Class similarity | Vocabulary | Number of images | Task |
|---|---|---|---|---|---|---|---|---|
| BDD100K [53] | General | Visible spectrum | Medium | 19 (Medium) | Low | Generic | 1,000 | Driving |
| Dark Zurich [41] | | Visible spectrum | Medium | 20 (Medium) | Low | Generic | 50 | Driving |
| MHP v1 [24] | | Visible spectrum | Small | 19 (Medium) | High | Task-spec. | 980 | Body parts |
| FoodSeg103 [47] | | Visible spectrum | Medium | 104 (Many) | High | Generic | 2,135 | Ingredients |
| ATLANTIS [11] | | Visible spectrum | Small | 56 (Many) | Low | Generic | 1,295 | Maritime |
| DRAM [8] | | Visible spectrum | Medium | 12 (Medium) | Low | Generic | 718 | Paintings |
| iSAID [46] | Earth Monitoring | Visible spectrum | Small | 16 (Medium) | Low | Generic | 4,055 | Objects |
| ISPRS Potsdam [4] | | Multispectral | Small | 6 (Few) | Low | Generic | 504 | Land use |
| WorldFloods [31] | | Multispectral | Medium | 3 (Binary) | Low | Generic | 160 | Floods |
| FloodNet [38] | | Visible spectrum | Medium | 10 (Few) | Low | Task-spec. | 5,571 | Floods |
| UAVid [29] | | Visible spectrum | Small | 8 (Few) | High | Task-spec. | 840 | Objects |
| Kvasir-Inst. [18] | Medical Sciences | Visible spectrum | Medium | 2 (Binary) | Low | Generic | 118 | Endoscopy |
| CHASE DB1 [13] | | Microscopic | Small | 2 (Binary) | Low | Domain-spec. | 20 | Retina scan |
| CryoNuSeg [30] | | Microscopic | Small | 2 (Binary) | Low | Domain-spec. | 30 | WSI |
| PAXRay-4 [42] | | Electromagnetic | Large | 4x2 (Binary) | Low | Domain-spec. | 180 | X-Ray |
| Corrosion CS [3] | Engineering | Visible spectrum | Medium | 4 (Few) | High | Task-spec. | 44 | Corrosion |
| DeepCrack [28] | | Visible spectrum | Small | 2 (Binary) | Low | Generic | 237 | Cracks |
| ZeroWaste-f [2] | | Visible spectrum | Medium | 5 (Few) | High | Generic | 929 | Conveyor |
| PST900 [43] | | Electromagnetic | Small | 5 (Few) | Low | Generic | 288 | Thermal |
| SUIM [17] | Agriculture and Biology | Visible spectrum | Medium | 8 (Few) | Low | Generic | 110 | Underwater |
| CUB-200 [45] | | Visible spectrum | Medium | 201 (Many) | High | Domain-spec. | 5,794 | Bird species |
| CWFID [15] | | Visible spectrum | Small | 3 (Few) | High | Generic | 21 | Crops |

## 3   MESS benchmark

Following the HELM benchmark [26] proposed for the evaluation of large language models, we develop a taxonomy with task characteristics for semantic segmentation and retrieve a set of more than 500 datasets that we review as part of the benchmark creation. For the development of the taxonomy, we use a method proposed by Nickerson et al. [35]. We start the development of the taxonomy by specifying the so-called meta-characteristic of the taxonomy (i.e., our goal): *identify visual and language characteristics of downstream tasks influencing the performance of zero-shot semantic segmentation models*. We then initialize the taxonomy in a conceptual-to-empirical cycle based on a review of other benchmarks and literature. Next, we refine the taxonomy in multiple empirical-to-conceptual iterations. We reviewed semantic segmentation datasets on Papers with Code, Kaggle, and additional test datasets used by recent segmentation models. We repeatedly reduced the dimensions of the taxonomy to the most meaningful ones for the meta-characteristic. We then conducted a statistical analysis of potential taxonomy dimensions to identify and remove similar or overlapping dimensions (see supplementary material). We identified multiple dimensions that highly correlate with each other like color map and sensor type, segment size and segments per image, as well as viewpoint and domain. Based on this analysis, we discarded color map, resolution, segments per image, and viewpoint. The final taxonomy matches all ending conditions [35]. While the proposed taxonomy identifies the most important dimensions and characteristics validated based on 120 classified datasets, there may be additional dimensions that influence the performance of zero-shot semantic segmentation models in specific cases. Overall, we observe that certain characteristics are more likely to co-occur. E.g, binary datasets typically imply a low class similarity, whereas task-specific vocabulary is often associated with a high similarity between the task-specific classes. We account for this imbalance in the distribution of the characteristics and reflect it in our benchmark.

Following the taxonomy development, we selected a representative set of datasets so that the MESS benchmark is informative, reproducible, and manageable. Specifically, we filtered the 120 classified datasets based on four exclusion criteria: each dataset has an official and annotated validation or test set, high annotation quality, moderate disk usage, and sufficient image size. Next, we selected a subset that consists of complementing use cases to avoid duplication and covers all characteristics of the taxonomy. We present the 22 selected datasets and their characteristics within the taxonomy's dimensions in Table 1 . These datasets cover a variety of applications, resulting in a holistic evaluation of domain-specific applications. We publish this new MESS benchmark at `https://blumenstiel.github.io/mess-benchmark` and invite others to suggest additional datasets and refine classes for future versions.

During dataset selection, we have not identified any ethical issues with these datasets based on the information provided by the data sources. Our use follows the terms and conditions set by the data providers, and we list the corresponding licenses in the supplementary material. However, we acknowledge the importance of considering the societal impact of our work. FMs, such as CLIP, are pre-trained on vast corpora of data that may contain biases. We refer to Agarwal et al. [1] for a detailed analysis of biases in CLIP. While the majority of MESS datasets are less prone to such biases, some may include data specific to gender or geographic regions. We believe that assessing model performance across a range of datasets can help to identify and mitigate the impact of biases.

## 4   Experimental setup

In this section, we provide a brief definition of the zero-shot semantic segmentation task, describe the metrics, and outline implementation details.

### 4.1   Task

Let $I$ denote an image with a set of candidate classes $\mathcal{C} = \{C_1, C_2, ..., C_N\}$, where each candidate class $C_i$ is described in natural language. Zero-shot semantic segmentation models then assign a class $C_i$ to each pixel of $I$. The number of candidate classes $N$ can vary during inference (e.g., different downstream tasks) and, additionally, the model may not have seen the candidate classes during training. This is in contrast to traditional semantic segmentation, where the set of classes is fixed during training and inference [7]. Each dataset represents a set of images with the same label set, and in our evaluations, none of the models is trained on the datasets from the benchmark

or the same set of candidate classes. However, it is reasonable to assume the evaluated classes have been present in the pre-training of the underlying vision-language models (like CLIP). All evaluated models have been trained on images with three channels (i.e., RGB). To account for datasets with varying numbers of input channels we mapped them to RGB (i.e., inputs with a single channel are mapped to RGB, for multispectral inputs we selected a subset of three channels).

## 4.2 Implementation

Following common practice, we evaluate all models using the mean of class-wise intersection over union (mIoU) [6, 7, 25, 50, 51]. We split very large images from the earth monitoring datasets into smaller patches of $1024 \times 1024$ pixels. Further, we use an IRRG color map for multispectral datasets (ISPRS Potsdam and WorldFloods) and select the thermal data in PST900. All other datasets include images with one or three channels. Across our implementation, we use PyTorch [36] and Detectron2 [48] for implementing the data loaders. For the convenience of users and contributors to our benchmark, we additionally provide wrappers for torchvision and MMSeg to process datasets in the Detectron2 dataset catalog. We did not train any models but used the publicly available weights and model configurations. The evaluation was conducted on an NVIDIA V100S.

## 4.3 Models

We utilize our MESS benchmark to evaluate a range of recent models for zero-shot semantic segmentation including the state-of-the-art, selecting models based on the reported performance and the availability of official code and weights. OVSeg [25], SAN [50], and CAT-Seg [7] represent the state-of-the-art across different approaches in the architecture for zero-shot semantic segmentation (i.e., two-stage mask-based, one-stage mask-based, and pixel-based). We additionally consider ZSSeg [51] and ZegFormer [9] which are frequently consulted as baseline models e.g. by [7, 25, 50]. The previously listed models are based on CLIP and use COCO Stuff to train the additional segmentation modules. Additionally, OVSeg uses COCO Captions for fine-tuning. X-Decoder [57] and OpenSeeD [55] are part of our evaluation since these approaches do not make use of CLIP but are based on UniCL [52] (i.e., their public versions). X-Decoder and OpenSeeD are trained on multiple datasets which we detail in the supplementary material.

To account for recent developments in the field, we additionally include SAM [20] in our evaluations. Standard SAM can only process visual prompts and does not facilitate text-to-mask settings. Therefore, we validated other ways to make use of SAM. We implement Grounded-SAM [16] using the predicted bounding boxes from Grounding DINO [27] as input for SAM and thereby enabling an open-vocabulary setting (i.e., text-to-mask). This serves as a baseline to better understand the potential of SAM-based text-to-mask models. The overall evaluation time per model on the MESS benchmark varies in our experiments between 1 hour for SAN-B and 14.5 hours for OVSeg-L.

# 5 Experiments

In the following, we provide a holistic overview of the performance of multiple zero-shot semantic segmentation models based on our MESS benchmark. We conduct a range of in-detail analyses of model performances across the dimensions of our taxonomy including sensor types, the class similarity, and the vocabulary—additional experiments are included in the supplementary material.

## 5.1 Multi-domain zero-shot semantic segmentation

We provide a quantitative comparison across models and all datasets summarized by their domain in Table 2 and per dataset results in Fig. 2 and 3. We add a random prediction as a lower bound by calculating the expected mIoU value with uniformly distributed predictions over all classes. In addition, we report fully supervised results based on the current SOTA from supervised semantic segmentation (see supplementary material). Overall, CAT-Seg-L achieves a strong performance across domains with an average mIoU of 38.14%, followed by its base and huge version. CAT-Seg is followed by SAN-L with a performance of 30.06%. Notably, the performance of zero-shot CAT-Seg-L in the general domain is only 8.69pp (average mIoU) below the performance of supervised SOTA approaches. In comparison, CAT-Seg-L reaches on average 50.36% of the supervised performance in earth monitoring and 54.18% on medical sciences. The performance gap compared to supervised

Table 2: mIoU results averaged by the dataset domain. Best-performing models are highlighted in bold, and the second-best are underlined. Random represents the randomly expected mIoU with uniformly distributed predictions. The best supervised models are separately selected for each dataset (see supplementary material for the supervised models and results).

| Model | Parameters | Inference (s/iter) | General | Earth Monit. | Medical Sciences | Engineer. | Agri. and Biology | Mean |
|---|---|---|---|---|---|---|---|---|
| *Random (LB)* | | | *1.17* | *7.11* | *29.51* | *11.71* | *6.14* | *10.27* |
| *Best supervised (UB)* | | | *48.62* | *79.12* | *89.49* | *67.66* | *81.94* | *70.99* |
| ZSSeg-B [51] | 211M | 0.49 | 19.98 | 17.98 | 41.82 | 14.0 | 22.32 | 22.73 |
| ZegFormer-B [9] | 210M | 0.18 | 13.57 | 17.25 | 17.47 | 17.92 | 25.78 | 17.57 |
| X-Decoder-T [57] | 164M | 0.1 | 22.01 | 18.92 | 23.28 | 15.31 | 18.17 | 19.8 |
| SAN-B [50] | 158M | 0.04 | 29.35 | 30.64 | 29.85 | 23.58 | 15.07 | 26.74 |
| OpenSeeD-T [55] | 116M | 0.08 | 22.49 | 25.11 | **44.44** | 16.5 | 10.35 | 24.33 |
| CAT-Seg-B [7] | 181M | 0.17 | **34.96** | **34.57** | 41.65 | **26.26** | **29.32** | **33.74** |
| OVSeg-L [25] | 531M | 1.64 | 29.54 | 29.04 | 31.9 | 14.16 | 28.64 | 26.94 |
| SAN-L [50] | 437M | 0.14 | 36.18 | 38.83 | 30.27 | 16.95 | 20.41 | 30.06 |
| CAT-Seg-L [7] | 490M | 0.33 | **39.93** | **39.85** | **48.49** | 26.04 | 34.06 | **38.14** |
| CAT-Seg-H [7] | 1049M | 0.5 | 37.98 | 37.74 | 34.65 | **29.04** | **37.76** | 35.66 |

models is even larger for the two other domains. Looking at the dataset-specific performance in Fig. 2 and 3, we observe that the performance varies between datasets and models. While SAN-L is the best-performing model on CUB-200 and DRAM, it has significantly lower performance on CWFID or CHASE DB1 compared to CAT-Seg-L. The model achieves scores between 50% and over 100% of the performance of supervised state-of-the-art in the general domain. Within the other domains, CAT-Seg-L has a performance gap of more than 25pp for most of the datasets.

The inference time varies between the models and, in particular, between different model architectures with some models requiring more than ten times higher computational effort indicated by higher inference times. In general, we observe the highest inference times for two-stage mask-based approaches, such as ZSSeg and OVSeg, which are between five to twelve times higher than other mask-based approaches (X-Decoder, OpenSeeD, and SAN). The point-based CAT-Seg uses a sliding window approach which requires five passes and therefore results in higher inference times than SAN. Overall, SAN represents the fastest model in our experiments.

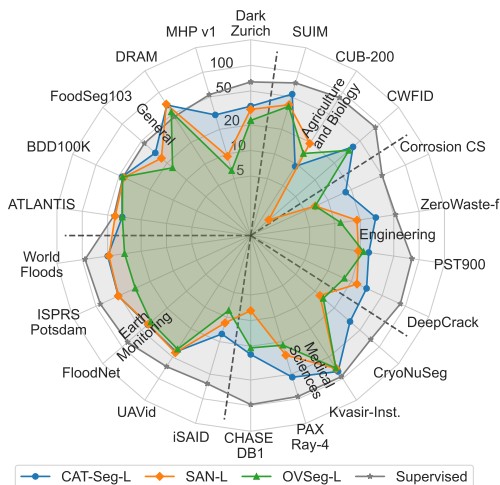

Figure 2: mIoU results for large models on a log scale. The datasets are grouped by their domain and sorted by supervised performance.

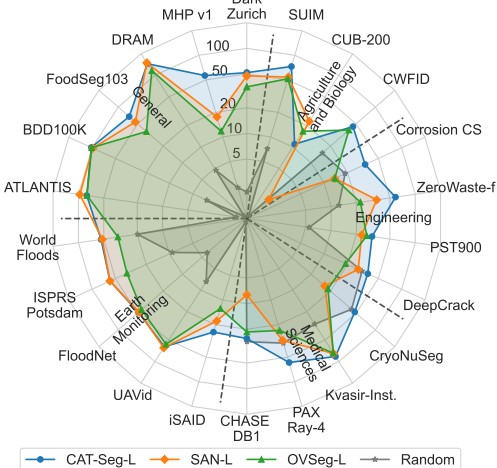

Figure 3: mIoU results of large models in relative comparison to the supervised mIoU on a log scale. 100 is equal to the supervised mIoU.

## 5.2 Sensor type evaluation

All considered models have been developed for the visual spectrum (i.e., RGB). In the following, we investigate the performance of three different sensor types: multispectral, electromagnetic, and microscopic. Three datasets from MESS allow for a direct comparison between different sensor types. For multispectral sensors, the MESS benchmark includes the IRRG color map for ISPRS Potsdam and WorldFloods. The models are able to process the different color maps and profit from the visual highlighting of vegetation through the infrared channel. This insight might be limited to commonly used color maps because other color maps might be less represented in the pre-training data of CLIP. On electromagnetic and thermal imagery, none of the evaluated models can regularly segment objects on the PST900 dataset. We compared this result to the aligned RGB images from PST900. All models perform significantly better on the RGB images. E.g., CAT-Seg-L reaches a mIoU of 65.55% on RGB images compared to only 25.26% for thermal data. We also tested a pseudo color map that maps the grayscale thermal data to a pseudo color scale, resulting in a similar low performance. Therefore, we conclude that zero-shot semantic segmentation models are currently not able to sufficiently segment objects in thermal images. Most models are also not able to correctly segment X-ray images in the PAXRay dataset, the second benchmark dataset with an electromagnetic sensor type. However, X-rays do include much more visual features compared to thermal images and CAT-Seg is able to segment some anatomical structures like the lungs. Further, the benchmark includes retina scans in CHASE DB1 and WSI images in CrypNuSeg to evaluate microscopic imagery. Similar to the PAXRay results, most models fail to segment the structures. But CAT-Seg and ZSSeg can locate the requested class. Thus, we assume that CLIP and zero-shot semantic segmentation can understand microscopic concepts but the correct segmentation is not achieved because of the small segments instead of the image type.

## 5.3 Multi-domain vs. in-domain evaluation

Most zero-shot semantic segmentation models are currently evaluated on five datasets: Pascal VOC, ADE20K-150, ADE20K-847, Pascal Context-59, and Pascal Context-459. Figure 4 compares the average results of the evaluated models on these common datasets (i.e., in-domain datasets) to a multi-domain setting with datasets of MESS benchmark. Note that the multi-domain datasets contain fewer classes on average, resulting in a much higher random mIoU. We provide the results for each dataset in the supplementary material. While SAN-L has comparable performance to the CAT-Seg models on common datasets, it has a significantly lower mIoU on domain datasets. Further, X-Decoder has a generally lower mIoU on domain datasets compared to other models. X-Decoder does not use CLIP which may explain the limited generalizability of the model. Overall, CAT-Seg is the only model architecture with a higher average mIoU on the domain datasets than common datasets.

## 5.4 Language characteristics

The differentiation between related classes is relevant in domain-specific use cases like biology. We analyze the influence of class similarity on class-wise IoU in Figure 5. Following Xu et al. [50], we calculated the class similarity as the maximum cosine similarity of the embedding to all other CLIP text embeddings in the label set. Overall, the class IoU does not correlate with the similarity. However, none of the classes with high similarity reaches a desirable IoU (e.g., the Corrosion CS dataset with three classes describing different corrosion stages). All models face difficulties in differentiating these classes. In additional experiments, the model performance significantly improved when considering similar classes as a single class. Also, specialized terms affect the model performance, specifically,

Table 3: Comparision of mIoU results for images with different sensor types. Pseudo refers to thermal data mapped to a pseudo color map.

| Model | ISPRS Potsdam | | WorldFloods | | PST900 | | |
| | IRRG | RGB | IRRG | RGB | Thermal | Pseudo | RGB |
| --- | --- | --- | --- | --- | --- | --- | --- |
| OVSeg-L [25] | 31.03 | 35.46 | 31.48 | 22.86 | 21.89 | 21.63 | 42.9 |
| SAN-L [50] | **51.45** | **52.06** | 48.24 | **45.93** | 19.01 | 19.41 | 49.02 |
| CAT-Seg-L [7] | 51.42 | 51.29 | **49.86** | 45.39 | **25.26** | **25.43** | **65.55** |

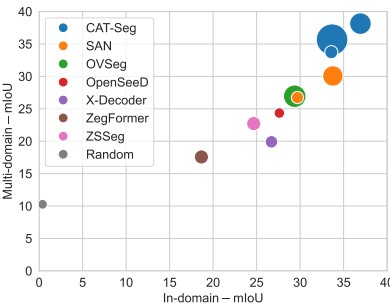

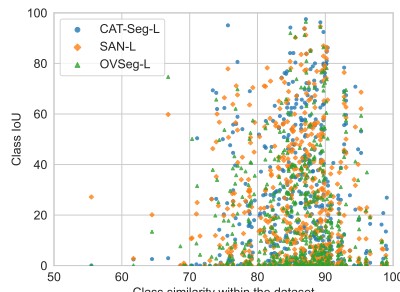

Figure 4: Cross domain settings can be challenging: Average mIoU of commonly used evaluation datasets in comparison to the results on the MESS benchmark. The size represents the parameter count of the models.

Figure 5: The class-wise IoU in comparison with the similarity to other labels within the dataset. The similarity is measured by the minimum cosine distance of the class label to all other CLIP text embeddings within the dataset.

domain-specific and task-specific labels. Our evaluation covers domain-specific words from medicine and biology, i.e., bird species and anatomical structures like the *mediastinum*. It shows that CLIP is able to understand domain-specific concepts to a limited extent. We observed higher performance for generic terminology. E.g., all models achieve higher performances on the Kvasir-Instrument dataset when using a generic vocabulary like *tool*. Utilizing a more precise term like *surgical instrument* reduces the mIoU. We refer to classes with specified conditions as task-specific classes. In our evaluations, CAT-Seg achieves the best results on task-specific classes. However, CAT-Seg still confuses classes and, e.g., predicts the right shoe and right leg significantly more often than the left side in MHP v1. CAT-Seg models are further biased towards the *parked car* class in UAVid images, while SAN and OVSeg mostly assign masks to the label *moving car*. Overall, domain-specific and task-specific vocabulary limits the performance of zero-shot semantic segmentation models.

## 5.5 Comparison to SAM

For a better understanding of current text-to-mask zero-shot semantic segmentation approaches, we compare them with grounded and oracle versions of SAM. SAM cannot directly process textual inputs, instead, it uses visual prompt inputs, i.e., bounding boxes or points. For the comparison, we implemented three versions of SAM. First, we made use of existing available demos combining Grounding DINO and SAM and extended them by a comprehensive quantitative evaluation. Second, oracle point-to-mask SAM refers to a model that provides a single point for every connected segment

Table 4: Domain-averaged mIoU results for Grounded-SAM and SAM with oracle inputs in a point-to-mask and box-to-mask setting. Random, supervised and CAT-Seg-L are provided for reference.

| Model | Input prompt | General | Earth Monitoring | Medical Sciences | Engineering | Agri. and Biology | Mean |
|---|---|---|---|---|---|---|---|
| *Random (LB)* | | *1.17* | *7.11* | *29.51* | *11.71* | *6.14* | *10.27* |
| *Best supervised (UB)* | | *48.62* | *79.12* | *89.49* | *67.66* | *81.94* | *70.99* |
| CAT-Seg-L [7] | | 39.93 | 39.85 | 48.49 | 26.04 | 34.06 | 38.14 |
| Gr.-SAM-B [16] | Grounding DINO [27] | 29.51 | 25.97 | 37.38 | **29.51** | 17.66 | 28.52 |
| Gr.-SAM-L [16] | | **30.32** | **26.44** | **38.69** | 29.25 | **17.73** | **29.05** |
| Gr.-SAM-H [16] | | 30.27 | **26.44** | 38.45 | 28.16 | 17.67 | 28.78 |
| SAM-B [20] | Oracle points [44] | **50.41** | 38.72 | 43.7 | 45.16 | 57.84 | 46.59 |
| SAM-L [20] | | 45.99 | **44.03** | 55.74 | **50.0** | **58.23** | **49.99** |
| SAM-H [20] | | 36.05 | 34.82 | **59.58** | 47.35 | 39.91 | 43.0 |
| SAM-B [20] | Oracle bounding boxes | **78.5** | **73.56** | 68.14 | **73.29** | 86.0 | **75.67** |
| SAM-L [20] | | 78.0 | 73.27 | 64.98 | 73.09 | **86.99** | 74.97 |
| SAM-H [20] | | 65.23 | 59.61 | 66.58 | 66.4 | 78.63 | 66.55 |

in the ground truth mask to simulate the visual input. We use the point sampling approach from RITM [44]. Third, oracle box-to-mask SAM utilizes a single box for every segment in the ground truth mask to simulate the visual input. We consider up to 100 input prompts per image to avoid a large number of very small segments. We later combine all predicted masks by taking the maximum logit value for each pixel. Pixels with only negative logit values are assigned to the background class or marked *unlabeled* in datasets without a background class. Note that inputting text data as in the models before is fundamentally different from utilizing visual inputs as in our two oracle SAM implementations and our analyses are not intended for a direct comparison but to better understand the potentials of SAM for zero-shot text-to-mask models.

In Table 4, we observe that the non-oracle implementation of SAM utilizing Grounding DINO generally exhibits limited performance compared to CAT-Seg text-to-mask models. Oracle versions of SAM receive significantly improved information on the location of the object and, therefore, show a strong performance. Given the perfect information on the location of objects in the image with oracle bounding boxes, the oracle box-to-mask SAM implementation even outperforms supervised semantic segmentation models. Overall, we observe that SAM models achieve a strong performance based on oracle information on the location of the objects. However text-to-mask zero-shot semantic segmentation models like CAT-Seg outperform the combination of Grounding DINO and SAM. Similar to X-Decoder and OpenSeeD, Grounding DINO does not use CLIP, which results in limited multi-domain performance. The results with oracle bounding boxes suggest that future combinations of SAM with open-vocabulary object detection models based on FMs like CLIP may outperform the current state-of-the-art in zero-shot semantic segmentation.

## 5.6 Qualitative analyses

In the following, we quantitatively compare the predictions of the three promising text-to-mask zero-shot semantic segmentation models with the ground truth and the grounding version of SAM on four different datasets (autonomous driving, satellite imagery, medical science, and engineering). We visually observe the following characteristics: First, CAT-Seg also visually surpasses the predictions of the other models. Second, across different domains, the predictions of CAT-Seg are largely in line with the ground truth and the segmentation is comparatively fine-grained. Third, we observe that Grounding DINO does not locate most segments and, therefore, Grounded-SAM tends to predict the background class. These qualitative observations are largely in line with our quantitative experiments.

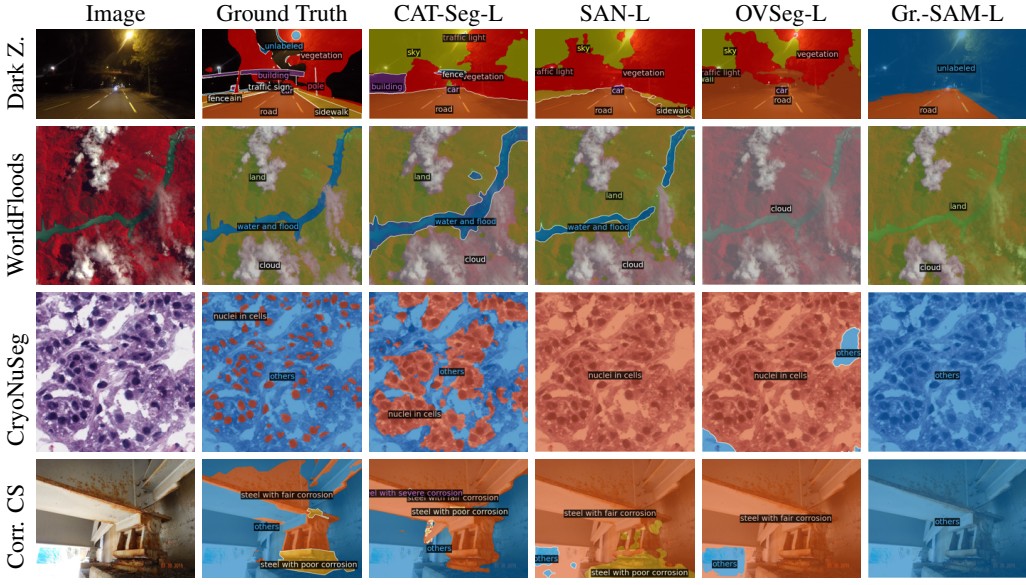

Figure 6: Predictions from selected datasets based on CAT-Seg-L [7], SAN-L [50], OVSeg-L [25], and Grounded-SAM [16].

Zero-shot semantic segmentation achieves a remarkable performance on in-domain datasets [7, 50]. Based on the MESS benchmark, we observe that these models can solve some tasks from other domains, however, are limited in their applicability to domains like medical science, engineering, and agriculture. We identified a range of challenges: First, we observe that domain-specific and task-specific vocabulary are difficult to handle. Models tend to be confused by labels with a high class similarity as in Corrosion CS. Therefore, we recommend to utilize a generic vocabulary with common class names, which led to improved performances in our experiments (e.g., *tool* instead of *medical instrument* in Kvasir-Instrument). Second, differences in the type of the sensor influence the performance of these models which are generally trained on the visual spectrum—for example, thermal data is hard to process. Third, we observe that state-of-the-art text-to-mask approaches outperform Grounded-SAM across multiple domains.

## 6    Conclusion

Zero-shot semantic segmentation has the potential to make segmentation models more accurate, cheap, flexible, and interactive. However, the current evaluation is limited to in-domain datasets, and previous analyses focused on model properties rather than task characteristics. With the MESS benchmark, we enable a holistic evaluation and invite others to utilize this benchmark to accelerate the field of semantic segmentation across domains to improve its real-world applicability.

## 7    Acknowledgements

We want to acknowledge the prior work this benchmark builds on. We especially want to emphasize that we leverage works across the AI community that should be recognized and cited. We appreciate the significant effort across the community in the careful collection, annotation, and publication of datasets. In the code repository, we provide additional details of the datasets including links to the corresponding works for citation. Additionally, we want to be explicit that our evaluation across diverse approaches would not be possible without publicly available architectures and corresponding model weights.

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
