# Supplementary Material
# What a MESS: Multi-Domain Evaluation of Zero-Shot Semantic Segmentation

**Benedikt Blumenstiel**\*      **Johannes Jakubik**\*      **Hilde Kühne**      **Michael Vössing**

The supplementary material is organized as follows:

- We detail the taxonomy development in Section A.
- The benchmark datasets are analyzed in Section B.
- We provide details about the evaluated models in Section C.
- Additional experiments are presented in Section D.
- Exemplary predictions are included in Section E.

## A   Taxonomy development

The taxonomy and the characterized datasets serve as a basis for the selection of the benchmark datasets. Therefore, we describe the taxonomy development in this section in detail. We applied the taxonomy development method proposed by Nickerson et al. [94] to analyze the task space of semantic segmentation. The method aims to develop a framework based on deduction and induction rather than *ad-hoc* decisions. We initialize the development by selecting our meta-characteristic (i.e., the goal): *identify visual and language characteristics of downstream tasks influencing the performance of zero-shot semantic segmentation models.*

We apply multiple empirical-to-conceptual or conceptual-to-empirical cycles until the ending conditions are reached. In an empirical-to-conceptual iteration, new objects are examined, and common characteristics are identified. The characteristics must derive from the meta-characteristic and discriminate between the objects to be of use for the taxonomy. In the conceptual step, the characteristics are grouped into dimensions. In contrast, a conceptual-to-empirical cycle starts by deducting potential dimensions and characteristics for the meta-characteristic based on prior knowledge. Next, the concept is evaluated by classifying objects. If a dimension does not differentiate between the objects or a characteristic has no real examples, it might not be appropriate. To fulfill the subjective ending conditions, the taxonomy must be concise, robust, comprehensive, extendible, and explanatory. Further, the objective ending conditions include, among others, dimension uniqueness and characteristic uniqueness within the dimension. We refer to Nickerson et al. [94] for more detailed information.

Starting with a first conceptual-to-empirical cycle, we analyzed other benchmarks and literature to initialize the taxonomy. The RF100 object detection benchmark [26] clusters the datasets into seven categories, representing different image types. We analyzed the literature of the RF100 categories to identify visual or language features relevant to these images. Aerial and satellite imagery has many important characteristics, such as the *sensor type* with different bands that can be mapped to true or false *color maps*, the *spatial resolution*, and *metadata* like time and location [97]. Electromagnetic images are often medical modalities using different sensor types, different *sensor directions* and having a domain-specific *vocabulary* that describes the anatomy [71]. Various sensors are also used in underwater imagery, and *preprocessing* plays an essential role in this image domain [81]. The spatial resolution is an important factor in microscopy, besides sensor types and specific hardware such as

---

\*Equal contributions

37th Conference on Neural Information Processing Systems (NeurIPS 2023) Track on Datasets and Benchmarks.

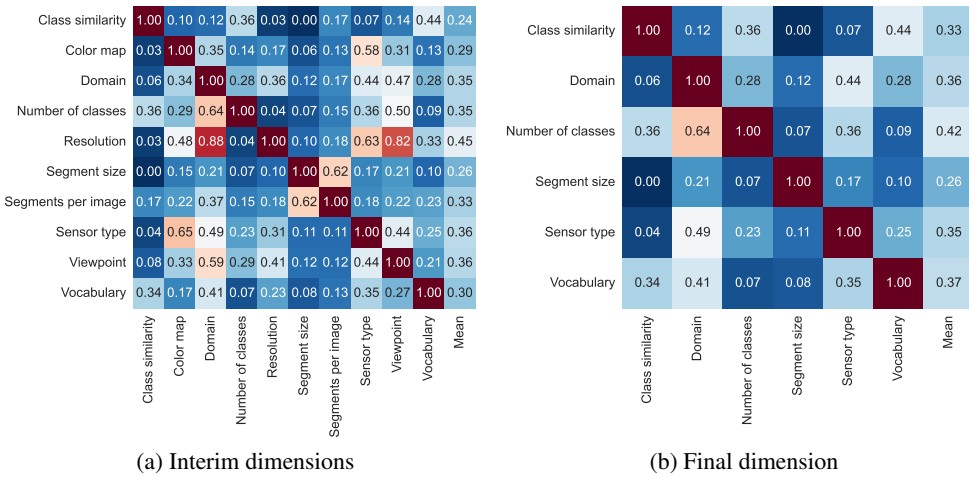

|                     | (a) Interim dimensions | (b) Final dimension |
|---|---|---|

Figure 1: Peason correlation between dimensions based on the classified datasets for an interim status (a) and the final taxonomy (b).

phase contrast or fluorescence [80]. Images from documents or video games are often synthetic but use the same visual spectrum as real-world images. Segmentation in documents does include very fine *segment sizes*. We further added a *domain* dimension because the image and labels are often domain-specific. Therefore, we select domains inspired by the major subject areas from Scopus[2] but shorten the labels to improve the usability of the taxonomy. We added a domain "General" for everyday images, which we did not associate with any subject area. Based on this research, we used all identified dimensions relevant to at least two domains as domain-specific dimensions lead to redundancy in the taxonomy. The initial characteristics of each dimension are selected based on the literature review and complemented through the following empirical step.

We refine the taxonomy in multiple empirical-to-conceptual iterations. Therefore, we reviewed overall 500 datasets, including all semantic segmentation datasets on Papers with Code[3], Kaggle[4], and test datasets from other segmentation models like SAM [66]. We classified 120 datasets within our taxonomy which are presented in Table 1. Note, that different versions of a dataset are classified if they lead to varying characteristics. We did not classify all datasets of similar use cases as we aim for a diverse collection of datasets (e.g., seven driving datasets out of 30+). Other criteria for exclusion are deviating tasks (e.g., 3D data) and missing data availability. Also, we discarded use cases that seem to be very unique, like galaxy segmentation. If only a few reviewed datasets covered specific use cases, e.g., crack segmentation, we analyzed additional datasets from other sources. Based on the datasets, we added dimensions regarding the segmentation mask and language-related dimensions like the *class similarity*. We repeatedly reduced the dimensions in conceptual phases to the most meaningful ones for the meta-characteristic.

Finally, we utilized a statistical analysis to identify similar dimensions, specifically, the Pearson correlation between each pair of dimensions using the empirical data from the classified datasets. We applied one-hot encoding for categorical dimensions and scaled each ordinal dimension by the number of characteristics. Figure 1 visualize multiple pairs with high correlation, e.g., *segment size* and *number of segments*. We reduced the interim dimensions based on the statistical analysis and the meta-characteristic. The final taxonomy passes all ending conditions in [94].

[2]List of subject areas: https://www.scopus.com/sources

[3]Semantic segmentation datasets: https://paperswithcode.com/datasets?task=semantic-segmentation

[4]Seach results for "semantic segmentation": https://www.kaggle.com/datasets?search=semantic+segmentation

Table 1: All 120 classified semantic segmentation datasets within the taxonomy.

| Dataset | Task | Domain | Sensor type | Segment size | Number of classes | Class similarity | Vocabulary |
|---|---|---|---|---|---|---|---|
| COCO Stuff [77] | Common | | Visible spectrum | Medium | 171 (Many) | Low | Generic |
| Pascal VOC 2012 [37] | Common | | Visible spectrum | Medium | 20 (Medium) | Low | Generic |
| ADE20K-150 [147] | Common | | Visible spectrum | Medium | 150 (Many) | Low | Generic |
| ADE20K-847 [147] | Common | | Visible spectrum | Medium | 847 (Many) | High | Generic |
| Pascal Context-59 [92] | Common | | Visible spectrum | Medium | 59 (Many) | Low | Generic |
| Pascal Context-459 [92] | Common | | Visible spectrum | Medium | 459 (Many) | High | Generic |
| LVIS [48] | Common | | Visible spectrum | Small | 1203 (Many) | High | Generic |
| FSS-1000 [74] | Common | | Visible spectrum | Large | 1000 (Many) | High | Generic |
| Mapillary Vistas v1 [93] | Driving | | Visible spectrum | Small | 66 (Many) | Low | Generic |
| Mapillary Vistas v2 [93] | Driving | | Visible spectrum | Small | 124 (Many) | Low | Task-spec. |
| Cityscapes [29] | Driving | | Visible spectrum | Small | 30 (Medium) | Low | Generic |
| BDD100K [140] | Driving | | Visible spectrum | Medium | 19 (Medium) | Low | Generic |
| Dark Zurich [105] | Driving | | Visible spectrum | Medium | 20 (Medium) | Low | Generic |
| SYNTHIA [102] | Driving | | Visible spectrum | Small | 13 (Medium) | Low | Generic |
| WoodScape [139] | Driving | | Visible spectrum | Small | 40 (Medium) | High | Generic |
| MVTec D2S [40] | Checkout | | Visible spectrum | Medium | 60 (Many) | Low | Generic |
| EgoHands [6] | Ego hands | | Visible spectrum | Medium | 5 (Few) | High | Task-spec. |
| WorkingHands [111] | Ego hands | General | Visible spectrum | Medium | 16 (Medium) | Low | Generic |
| EgoHOS [143] | Ego hands | | Visible spectrum | Medium | 8 (Few) | High | Task-spec. |
| EYTH [124] | Ego hands | | Visible spectrum | Medium | 2 (Binary) | Low | Generic |
| VISOR [30] | Ego hands | | Visible spectrum | Small | 257 (Many) | High | Generic |
| Open Surfaces [9] | Materials | | Visible spectrum | Medium | 37 (Medium) | High | Domain-spec. |
| MINC [10] | Materials | | Visible spectrum | Medium | 23 (Medium) | Low | Generic |
| DMS [123] | Materials | | Visible spectrum | Small | 52 (Many) | High | Generic |
| DeepFashion2 [45] | Clothing | | Visible spectrum | Small | 13 (Medium) | Low | Generic |
| ModaNet [146] | Clothing | | Visible spectrum | Small | 13 (Medium) | Low | Generic |
| MHP v1 [72] | Body parts | | Visible spectrum | Small | 18 (Medium) | High | Task-spec. |
| MHP v2 [72] | Body parts | | Visible spectrum | Small | 58 (Many) | High | Task-spec. |
| FoodSeg103 [135] | Ingredients | | Visible spectrum | Medium | 103 (Many) | High | Generic |
| TACO [98] | Trash | | Visible spectrum | Medium | 60 (Many) | High | Domain-spec. |
| RailSem19 [141] | Rail | | Visible spectrum | Small | 11 (Medium) | High | Task-spec. |
| ATLANTIS [36] | Maritime | | Visible spectrum | Small | 56 (Many) | Low | Generic |
| Aircraft Context [115] | Aerial vehicles | | Visible spectrum | Medium | 8 (Few) | Low | Generic |
| RELLIS-3D [64] | Robotics | | Visible spectrum | Small | 20 (Medium) | Low | Generic |
| SketchyScene-7k [148] | Sketches | | Visible spectrum | Small | 45 (Medium) | Low | Generic |
| DRAM [28] | Paintings | | Visible spectrum | Medium | 12 (Medium) | Low | Generic |
| iSAID [134] | Objects | | Visible spectrum | Small | 15 (Medium) | Low | Generic |
| DSTL Satellite [59] | Objects | | Multispectral | Small | 10 (Medium) | High | Generic |
| ISPRS Potsdam [16] | Land use | | Multispectral | Small | 6 (Few) | Low | Generic |
| LandCoverNet [4] | Land use | | Multispectral | Medium | 7 (Few) | Low | Generic |
| LoveDA [129] | Land use | | Visible spectrum | Small | 7 (Few) | Low | Generic |
| Deep Globe [32] | Land use | | Visible spectrum | Medium | 7 (Few) | Low | Generic |
| GID-5 [120] | Land use | | Multispectral | Small | 5 (Few) | Low | Generic |
| GID-15 [120] | Land use | | Multispectral | Small | 16 (Medium) | High | Task-spec. |
| Dubai [57] | Land use | | Visible spectrum | Small | 6 (Few) | Low | Generic |
| Sen1Floods11 [13] | Floods | | Electromagnetic | Small | 2 (Binary) | Low | Generic |
| WorldFloods [84] | Floods | | Multispectral | Medium | 3 (Binary) | Low | Generic |
| HR-GLDD [87] | Landslides | | Multispectral | Medium | 2 (Binary) | Low | Generic |
| Antarctic fracture [67] | Ice fractures | Earth | Multispectral | Small | 2 (Binary) | Low | Generic |
| Active fire [31] | Wildfires | Monitoring | Multispectral | Small | 2 (Binary) | Low | Generic |
| xBD [49] | Buildings | | Visible spectrum | Small | 5 (Few) | High | Task-spec. |
| MSAW [110] | Buildings | | Electromagnetic | Small | 2 (Binary) | Low | Generic |
| 3D PV Locator [85] | PV | | Visible spectrum | Small | 2 (Binary) | Low | Generic |
| AgricultureVision [24] | Agriculture | | Multispectral | Medium | 9 (Few) | Low | Domain-spec. |
| PASTIS [44] | Agriculture | | Multispectral | Small | 18 (Medium) | High | Domain-spec. |
| CalCROP21 [47] | Agriculture | | Multispectral | Small | 29 (Medium) | High | Domain-spec. |
| Arctic Sea Ice [117] | Sea ice | | Multispectral | Medium | 8 (Few) | High | Task-spec. |
| ELAI Dust Storm [7] | Dust storm | | Visible spectrum | Large | 2 (Binary) | Low | Generic |
| FloodNet [101] | Floods | | Visible spectrum | Medium | 10 (Medium) | Low | Task-spec. |
| SDD [60] | Objects | | Visible spectrum | Small | 21 (Medium) | Low | Generic |
| UDD [22] | Objects | | Visible spectrum | Medium | 6 (Few) | Low | Generic |
| UAVid [82] | Objects | | Visible spectrum | Small | 6 (Few) | High | Task-spec. |
| PV thermography [132] | PV | | Electromagnetic | Small | 6 (Binary) | High | Domain-spec. |
| CholecSeg8k [122] | Surgery | | Visible spectrum | Medium | 13 (Medium) | High | Domain-spec. |
| RoboTool [43] | Surgery | | Visible spectrum | Medium | 2 (Binary) | Low | Generic |
| Kvasir-Instrument [62] | Surgery | | Visible spectrum | Medium | 2 (Binary) | Low | Generic |
| ROBUST-MIS 2019 [103] | Surgery | | Visible spectrum | Medium | 2 (Binary) | Low | Generic |
| Kvasir SEG [63] | Surgery | Medical | Visible spectrum | Medium | 2 (Binary) | Low | Domain-spec. |
| Vocalfolds [68] | Surgery | Sciences | Visible spectrum | Medium | 7 (Few) | Low | Domain-spec. |
| CHASE DB1 [41] | Retina scan | | Microscopic | Small | 2 (Binary) | Low | Domain-spec. |
| HRF [69] | Retina scan | | Microscopic | Small | 2 (Binary) | Low | Domain-spec. |
| STARE [55] | Retina scan | | Microscopic | Small | 2 (Binary) | Low | Domain-spec. |
| Intraretinal C. Fluid [2] | Retinal OCT | | Microscopic | Small | 2 (Binary) | Low | Domain-spec. |

| Dataset | Task | Domain | Sensor type | Segment size | Number of classes | Class similarity | Vocabulary |
|---|---|---|---|---|---|---|---|
| GLaS [113] | WSI | | Microscopic | Medium | 2 (Binary) | High | Domain-spec. |
| Gleason [95] | WSI | | Microscopic | Large | 6 (Few) | High | Domain-spec. |
| CryoNuSeg [83] | WSI | | Microscopic | Small | 2 (Binary) | Low | Domain-spec. |
| BBBC038v1 [17] | WSI | | Microscopic | Small | 2 (Binary) | Low | Domain-spec. |
| Vector-LabPics [35] | Lab vessels | | Visible spectrum | Medium | 58 (Medium) | High | Domain-spec. |
| vesselNN [118] | Brain vessel | | Microscopic | Small | 2 (Binary) | Low | Domain-spec. |
| MTNeuro [99] | Brain vessel | Medical | Microscopic | Small | 3 (Few) | High | Domain-spec. |
| Neuronal Cells [53] | Brain cells | Sciences | Microscopic | Small | 2 (Binary) | Low | Domain-spec. |
| BraTS 2015 [88] | Brain tumor | | Electromagnetic | Medium | 5 (Few) | High | Domain-spec. |
| ISIC2018 Task1 [27] | Lesions | | Visible spectrum | Large | 2 (Binary) | Low | Domain-spec. |
| PAXRay-166 [108] | X-Ray | | Electromagnetic | Small | 166x2 (Binary) | High | Domain-spec. |
| PAXRay-4 [108] | X-Ray | | Electromagnetic | Large | 4x2 (Binary) | Low | Domain-spec. |
| Pulmonary Chest [18] | X-Ray | | Electromagnetic | Large | 2 (Binary) | Low | Generic |
| US segmentation [125] | Ultrasound | | Electromagnetic | Medium | 9 (Few) | High | Domain-spec. |
| Severstal [109] | Surface defect | | Visible spectrum | Medium | 4 (Few) | High | Domain-spec. |
| KolektorSDD2 [14] | Surface defect | | Visible spectrum | Medium | 2 (Binary) | Low | Generic |
| EMPS [138] | Particles | | Electromagnetic | Small | 2 (Binary) | Low | Generic |
| LIB-HSI [50] | Building fasade | | Multispectral | Medium | 44 (Medium) | High | Generic |
| Corrosion CS [11] | Corrosion | | Visible spectrum | Medium | 4 (Few) | High | Task-spec. |
| LCW [12] | Cracks | | Visible spectrum | Small | 2 (Binary) | Low | Generic |
| DeepCrack [79] | Cracks | | Visible spectrum | Small | 2 (Binary) | Low | Generic |
| ZeroWaste-f [8] | Conveyor | Engineering | Visible spectrum | Medium | 4 (Few) | High | Generic |
| Thermal Dog [104] | Thermal | | Electromagnetic | Medium | 3 (Few) | Low | Generic |
| PST900 [112] | Thermal | | Electromagnetic | Small | 5 (Few) | Low | Generic |
| TAS-NIR [91] | Thermal | | Electromagnetic | Medium | 22 (Medium) | High | Generic |
| PIDRay [127] | Security | | Electromagnetic | Small | 12 (Medium) | Low | Generic |
| TTPLA [1] | Powerlines | | Visible spectrum | Small | 5 (Few) | High | Generic |
| Vale [56] | Terrain | | Visible spectrum | Medium | 5 (Few) | High | Task-spec. |
| AI4MARS [116] | Terrain | | Visible spectrum | Small | 4 (Few) | High | Generic |
| TrashCan [54] | Trash | | Visible spectrum | Medium | 4 (Few) | Low | Generic |
| SUIM [61] | Underwater | | Visible spectrum | Medium | 8 (Few) | Low | Generic |
| DeepFish [106] | Fish | | Visible spectrum | Medium | 2 (Binary) | Low | Generic |
| NDD20 [121] | Fish | | Visible spectrum | Medium | 2 (Binary) | Low | Generic |
| Ciona17 [42] | Maritime species | | Visible spectrum | Large | 4 (Few) | High | Domain-spec. |
| CUB-200 [126] | Bird species | | Visible spectrum | Medium | 201 (Many) | High | Domain-spec. |
| Oxford-IIIT Pet [96] | Animal species | | Visible spectrum | Large | 28 (Medium) | High | Domain-spec. |
| Plittersdorf [51] | Animals | | Electromagnetic | Medium | 2 (Binary) | Low | Generic |
| CAMO [70] | Animals | Agriculture | Visible spectrum | Medium | 2 (Medium) | Low | Domain-spec. |
| COD [38] | Animals | and Biology | Visible spectrum | Medium | 78 (Many) | Low | Domain-spec. |
| CropAndWeed [114] | Plants | | Visible spectrum | Small | 100 (Many) | High | Domain-spec. |
| WGISD [107] | Plants | | Visible spectrum | Medium | 2 (Binary) | Low | Generic |
| PPDPS [90] | Plants | | Visible spectrum | Large | 2 (Binary) | Low | Generic |
| Plant seg. [34] | Plants | | Visible spectrum | Small | 3 (Few) | High | Task-spec. |
| CWFID [52] | Crops | | Visible spectrum | Small | 3 (Few) | High | Generic |
| PPDLS [90] | Leefs | | Visible spectrum | Medium | 2 (Binary) | Low | Generic |
| Leaf disease [3] | Leef disease | | Visible spectrum | Small | 2 (Binary) | Low | Generic |
| Rice Leaf dis. [75] | Leef disease | | Visible spectrum | Small | 5 (Few) | High | Domain-spec. |

# B  Benchmark datasets

## B.1  Overview

We selected 22 out of the 120 classified datasets for the MESS benchmark. The links, licenses, selected splits, and a sample of the class labels of the datasets are provided in Table 2. We specified some label names for better performances of the models similar to [73]. E.g., we use *crop seedling* instead of *crop* for the CWFID dataset. We refer to our implementation for all class labels.

We shortly introduce each dataset in the following: The general datasets include datasets with everyday scenes but more specific use cases and niche image themes in comparison to the standard evaluation datasets. Specifically, the use cases include two driving datasets with one covering nighttime images. Further, MHP v1 covers classes of body parts and clothes while FoodSeg103 requires the segmentation of different ingredients. The ATLANTIS dataset focuses on classes related to maritime environments and DRAM covers common classes in paintings. The selected earth monitoring datasets include iSAID, which requires the segmentation of 15 object categories in satellite images, e.g., a tennis court or a helicopter. ISPRS Potsdam and WorldFloods provide multispectral data, and our main evaluation uses an IRRG false color mapping. Near-infrared radiation is visualized in red and highlights vegetation. ISPRS Potsdam provides very high-resolution images of an urban area with multiple classes, while WorldFloods has a 10-meter resolution and focuses on

Table 2: Details for the 22 MESS datasets including the links and licenses. Nearly all datasets require attribution and many only allow non-commercial use.

| Dataset | Link | Licence | Split | No. of classes | Classes |
|---|---|---|---|---|---|
| BDD100K [140] | berkeley.edu | custom | val | 19 | [road; sidewalk; building; wall; fence; pole; traffic light; traffic sign; ...] |
| Dark Zurich [105] | ethz.ch | custom | val | 20 | [unlabeled; road; sidewalk; building; wall; fence; pole; traffic light; ...] |
| MHP v1 [72] | github.com | custom | test | 19 | [others; hat; hair; sunglasses; upper clothes; skirt; pants; dress; ...] |
| FoodSeg103 [135] | github.io | Apache 2.0 | test | 104 | [background; candy; egg tart; french fries; chocolate; biscuit; popcorn; ...] |
| ATLANTIS [36] | github.com | Flickr (images) | test | 56 | [bicycle; boat; breakwater; bridge; building; bus; canal; car; ...] |
| DRAM [28] | ac.il | custom (in download) | test | 12 | [bird; boat; bottle; cat; chair; cow; dog; horse; ...] |
| iSAID [134] | github.io | Google Earth (images) | val | 16 | [others; boat; storage tank; baseball diamond; tennis court; bridge; ...] |
| ISPRS Potsdam [16] | isprs.org | no licence provided[a] | test | 6 | [road; building; grass; tree; car; others] |
| WorldFloods [84] | github.com | CC NC 4.0 | test | 3 | [land; water and flood; cloud] |
| FloodNet [101] | github.com | custom | test | 10 | [building-flooded; building-non-flooded; road-flooded; water; tree; ...] |
| UAVid [82] | uavid.nl | CC BY-NC-SA 4.0 | val | 8 | [others; building; road; tree; grass; moving car; parked car; humans] |
| Kvasir-Inst. [62] | simula.no | custom | test | 2 | [others; tool] |
| CHASE DB1 [41] | kingston.ac.uk | CC BY 4.0 | test | 2 | [others; blood vessels] |
| CryoNuSeg [83] | kaggle.com | CC BY-NC-SA 4.0 | test | 2 | [others; nuclei in cells] |
| PAXRay-4 [108] | github.io | custom | test | 4x2 | [others, lungs], [others, bones], [others, mediastinum], [others, diaphragm] |
| Corrosion CS [11] | figshare.com | CC0 | test | 4 | [others; steel with fair corrosion; ... poor corrosion; ... severe corrosion] |
| DeepCrack [79] | github.com | custom | test | 2 | [concrete or asphalt; crack] |
| PST900 [112] | github.com | GPL-3.0 | test | 5 | [background; fire extinguisher; backpack; drill; human] |
| ZeroWaste-f [8] | ai.bu.edu | CC-BY-NC 4.0 | test | 5 | [background or trash; rigid plastic; cardboard; metal; soft plastic] |
| SUIM [61] | umn.edu | MIT | test | 8 | [human diver; reefs and invertebrates; fish and vertebrates; ...] |
| CUB-200 [126] | caltech.edu | custom | test | 201 | [background; Laysan Albatross; Sooty Albatross; Crested Auklet; ...] |
| CWFID [52] | github.com | custom | test | 3 | [ground; crop seedling; weed] |

[a]Upon request, the naming of the data provider and project is required.

water segmentation. We selected two drone datasets with similar use cases. UAVid includes urban scenes, and FloodNet covers flooded buildings and roads. The medical datasets cover four different modalities: Endoscopy (RGB images), retinal scans, whole slide imagery (WSI), and X-ray scans. Each binary segmentation task focuses on a specific object or anatomical structure, like blood vessels or lungs. The multi-label segmentation dataset PAXRay is a special case. We do not use each of the 166 annotated classes but only the four superclasses. Because of the mask overlay, each class is predicted in a binary setting, and we average the resulting metrics. Next, we selected four diverse engineering datasets. Corrosion CS includes images of corrosion on bridges and other infrastructure with four different condition states. DeepCrack consists of close-up images of crack. PST900 consists of thermal imagery with firefighter-related objects. We use a gray-scale color map in our main evaluation to visualize the thermal data. The Zero-Waste-f dataset includes images of a conveyor belt with annotations for four types of recyclable trash. The final three datasets cover biological-related datasets: SUIM is an underwater imagery dataset with fish, aquatic plants, and others. CUB-200 is a widely used dataset of 200 bird species. The images of CUB-200 are relatively easy to segment, but assigning the correct species is challenging. CWFID includes crop seedlings and weeds.

We looked up the current fully supervised performance to provide an upper threshold for each dataset and present them in Table 3. We did not find any mIoU results for the MHP v1 dataset as it is originally annotated for instance segmentation. Therefore, we trained MaskFormer [23] to provide a reference. We trained the model for 100K steps using the Swin-B ADE20K-150 settings and evaluated the best model based on the val mIoU.

## B.2 Dataset analysis

The classified datasets are visualized in Figure 2 by applying a Principal Component Analysis (PCA) along the taxonomy's dimensions. An analysis of the principal components reveals that, apart from the domain, mainly language-related features differentiate the datasets within the taxonomy. The PCA has two big clusters covering all domains – one cluster of datasets (top) with mostly domain-specific vocabulary and high class similarity and another one (bottom) with tasks of easily distinguishable generic classes. The PCA emphasizes the importance of these two dimensions for all domains. The datasets visualized in the center between these clusters have either a domain-specific vocabulary with low class similarly, which is often the case for medical datasets, or the opposite, often observed in

Table 3: Supervised mIoU results for the datasets.

| Dataset | Model | Year | mIoU |
|---------|-------|------|------|
| BDD100K | Two-branch Enet [89] | 2023 | 44.8 |
| Dark Zurich | Refign (HRDA) [15] | 2023 | 63.9 |
| MHP v1 | MaskFormer (Swin-B) [23] | 2021 | 53.18$^a$ |
| FoodSeg103 | SeTR-MLA (ViT-16/B) [145] | 2021 | 45.1 |
| ATLANTIS | AQUANet [36] | 2021 | 42.22 |
| DRAM | DRAM model [28] | 2022 | 45.71$^b$ |
| iSAID | IMP-ViTAEv2-S-UperNet [128] | 2022 | 65.3 |
| ISPRS Potsdam | DC-Swin [130] | 2022 | 87.56 |
| WorldFloods | UNet [84] | 2021 | 92.71 |
| FloodNet | SegFormer [5] | 2023 | 82.22 |
| UAVid | UNetFormer [131] | 2022 | 67.8 |
| Kvasir-Instrument | U-Net [62] | 2021 | 93.7 |
| CHASE DB1 | RV-GAN [65] | 2021 | 97.05 |
| CryoNuSeg | TransUNet [21] | 2022 | 73.45 |
| PAXRay-4 | Unet-R50 [108] | 2022 | 93.77 |
| Corrosion CS | DeeplabV3+ [11] | 2021 | 49.92 |
| DeepCrack | DeepCrack-GF [79] | 2019 | 85.9 |
| ZeroWaste-f | DeeplabV3+ [8] | 2022 | 52.5 |
| PST900 | SpiderMesh [39] | 2022 | 82.3 |
| SUIM | LOCA [19] | 2022 | 74.0 |
| CUB-200 | GFN [144] | 2022 | 84.6 |
| CWFID | Unet-Resnet-50 [119] | 2022 | 87.23 |

$^a$Own experiment because mIoU results are not reported in MHP v1 literature.
$^b$The DRAM model is not trained on a labeled training set but self-supervised on generated images.

general datasets. Furthermore, medical datasets have few classes, while general use cases have many classes, with the other three domains in between.

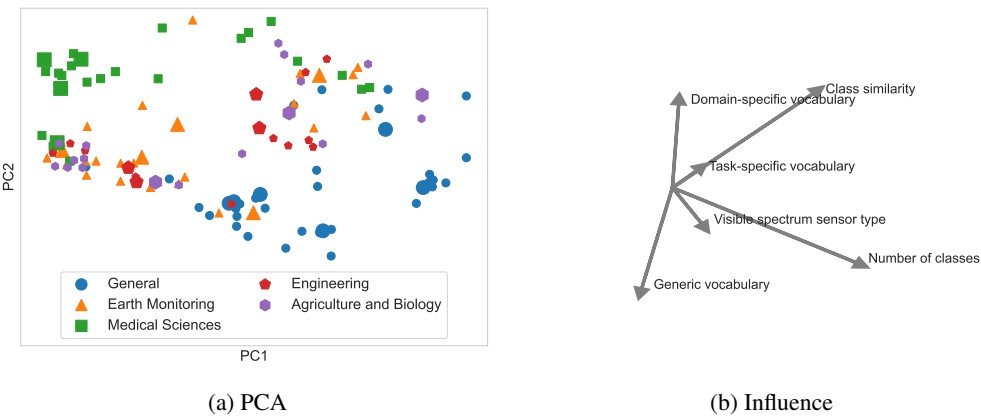

(a) PCA                                        (b) Influence

Figure 2: PCA of the classified datasets, clustered by their domain (a), and the highest influencing factors apart from the domains (b). An increased size visualizes selected datasets. Some noise was added to visualize similar classified datasets.

Following Xu et al. [136], we conducted an analysis of the similarity between the labels of each dataset and the training labels from COCO-Stuff [77] which is used by most evaluated models. The similarity between two labels is computed using the cosine similarity between their CLIP text embeddings. Next, we select the maximum similarity value for each text label (i.e., the minimal distance of this label to the training labels). To calculate the similarity between a test set and the training set, we can select the minimum value among the test labels. This represents a Hausdoff

Distance between these two sets, i.e., the maximum distance in the embedding space [136]. However, this calculation is sensitive to outliers and we also report the mean similarity over all test labels.

The analysis in Figure 3 visualizes that most datasets do include classes with a low train similarity that are not related to the train labels. Some datasets have a high mean similarity (i.a., BDD100K, DRAM, ISPRS Potsdam, ZeroWaste-f). Therefore, most classes in these datasets are equal or similar to a training label from COCO-Stuff. The medical and engineering datasets often have a low mean train similarity and include labels that are not present in the training labels.

Additionally, Figure 3 includes the similarity values within each dataset. The similarity is calculated using the maximum cosine similarity for each label to the rest. Selecting the maximum value from all labels results in the inner max similarity, and a high value indicates that at least two labels in the task have very close embeddings. It corresponds to a high class similarity within our taxonomy. Therefore, these classes are challenging to differentiate, even without considering the image features (e.g., classes in MHP v1, Corrosion CS, and CUB-200).

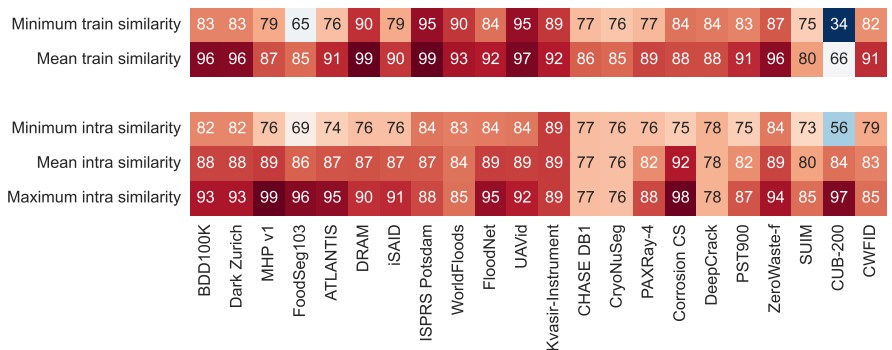

Figure 3: Class similarity to the COCO-Stuff training labels and within each dataset.

## C   Models

We provide an overview of the tested zero-shot semantic segmentation models in Table 4 including their modules and training datasets. We only include the datasets used for training the segmentation model and not the pre-training datasets of a utilized FM. We want to point out that the public versions of X-Decoder and OpenSeeD are using different FMs than the larger, non-public versions.

We utilize Grounded-SAM based on a re-implementation inspired by the demo code in [58]. To our knowledge, other implementations of Grounded-SAM are limited to demo scripts and do not apply semantic segmentation. The model combines bounding box predictions from Grounding DINO [78] with instance segmentations from SAM [66]. Grounding DINO is an open-vocabulary object detection model. The model predicts bounding boxes for all class labels in the label set. The labels also include the background class, as we noticed better results in prior experiments compared to discarding the background class. Next, SAM predicts one instance segmentation mask for each bounding box, and the pixel-wise confidence values are scaled by the confidence score of the bounding box. The instance masks of each class are combined by the maximum confidence value of each pixel, resulting in semantic masks. Negative values represent background predictions. Therefore, pixels with only negative values are predicted as background or no-object for datasets without a background class.

We noticed that Grounding DINO has a limited capability to predict non-general classes. SAM can also be combined with other open-vocabulary object detection models to improve performance. We refer to the oracle bounding box results for an upper bound.

## D   Additional results

We provide further experiments and detailed dataset-wise results in this section. Specifically, we analyze classes of interest, the similarity between evaluation and training classes, and the influence of the segment size.

Table 4: Overview of the evaluated models.

| Name | Versions | Year | Modules | Training datasets |
|---|---|---|---|---|
| ZSSeg [137] | Base | 2021 | CLIP ViT-B & text encoder, Resnet 101 | COCO-Stuff |
| ZegFormer [33] | Base | 2022 | CLIP ViT-B & text encoder, Resnet 101 | COCO-Stuff-156 |
| OVSeg [76] | Large | 2022 | CLIP ViT-L & text encoder, Swin-B | COCO-Stuff, COCO Caption |
| X-Decoder [149] | Tiny | 2023 | Focal-T/L, UniCL text encoder, ViT-decoder | COCO2017, 4M corpora |
| OpenSeeD [142] | Tiny | 2023 | Swin-T, UniCL text encoder, ViT-decoder | COCO2017, Objects365 |
| SAN [136] | B/L | 2023 | CLIP ViT-B/L & text encoder, ViT-adapter | COCO-Stuff |
| CAT-Seg [25] | B/L/H | 2023 | CLIP ViT-B/L/H & text encoder, Swin-B, ViT-decoder | COCO-Stuff |
| Grounded-SAM [58] | B/L/H | 2023 | SAM ViT-B/L/H, DINO Swin-B, BERT-B | SA-1B, COCO, O365, GoldG, Cap4M, OpenImage, ODinW-35, RefCOCO |

## D.1 Classes of interest

For several datasets in the benchmark, a subset of the annotated classes is particularly relevant. We refer to this subset as Class(es) of Interest (CoI). E.g., binary segmentation tasks typically include a CoI (like pixels depicting a flood event) and a background class. In many cases, the model performance varies between CoI and background. To better understand the actual performance for these classes, we report the mIoU on the CoI subset (CoI-mIoU). With $CoI \subseteq C$, and $IoU_i$ being the intersection over union for class $i$, we calculate the metric $CoI\text{-}mIoU = \frac{\sum_{i \in CoI} IoU_i}{|CoI|}$. In binary segmentation tasks, this is similar to the $IoU_{pos}$ of the positive class [86].

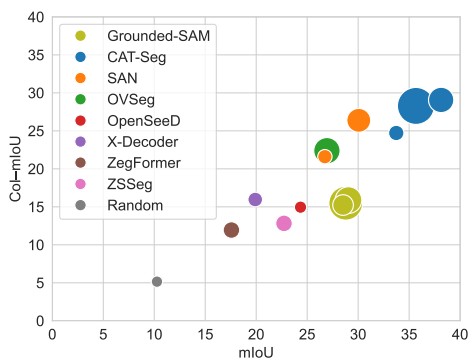

Figure 4: mIoU for Class(es) of Interest (CoI) in comparison to the mIoU of all classes. The size represents the parameter count of the models.

Figure 4 visualizes the CoI-mIoU compared to the mIoU of all predicted classes. On average, none of the models is able to segment classes of interest as well as all classes. Models like ZSSeg, OpenSeeD, and Grounded-SAM have a particularly strong bias toward misclassifying CoI than the other models. Also, CAT-Seg tends to misclassify classes of interest. For example, SAN-L has on average only a 3.18pp lower CoI-mIoU than the best-performing model CAT-Seg-L while the mIoU difference is 8.08pp. The differences between the mIoU and the CoI-mIoU vary between the datasets and domains. Figure 5 visualizes the mIoU and CoI-mIoU for all datasets and model architectures. The differences between both class sets are most evident for medical, engineering, and biological datasets (except for the datasets Kvasir-Instrument and SUIM). The CoI seem to be challenging for all models. These classes often include characteristics such as small segments, a high class similarity, or domain-specific labels. Furthermore, several model architectures tend to predict no or very few pixels as CoI, resulting in very low or zero scores. These architectures include X-Decoder, OpenSeeD, and Grounded-SAM. The models do not make use of CLIP, which may limit their capability to generalize to the domain-specific classes.

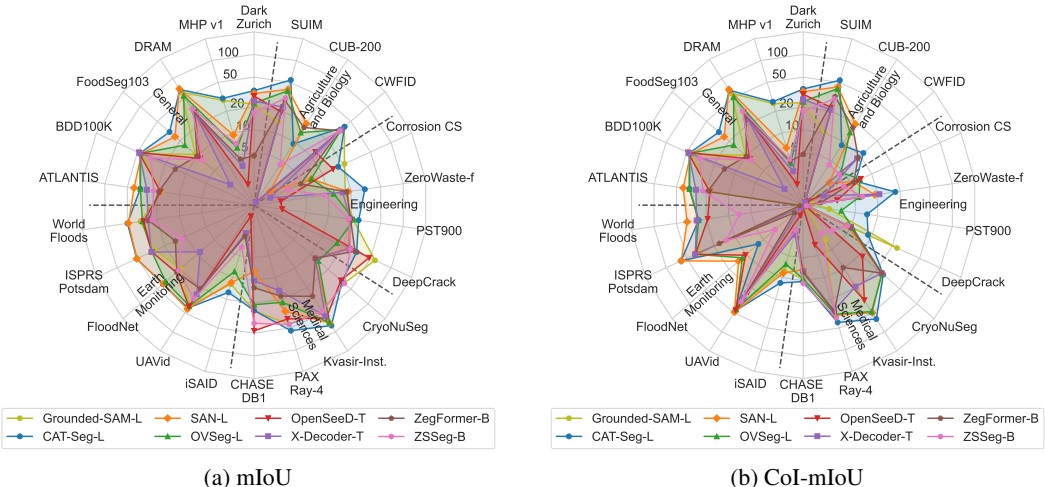

(a) mIoU                                    (b) CoI-mIoU

Figure 5: mIoU (a) and CoI-mIoU (b) results for all model architectures on a log scale.

## D.2 Similarity to training classes

The generalized zero-shot transfer setting does allow an overlap between the training labels and the evaluation labels. We analyze this overlap and the influences on the model performance by calculating the embedding similarity of each label to the training labels in COCO-Stuff. A high similarity corresponds to the concept being present in the training dataset. Figure 6 presents the correlation between the similarity and the class-wise IoU for the three large models which are trained on COCO-Stuff. The results indicate a positive correlation between the similarity of the training labels and the performance. We also observe a comparable correlation for all other model architectures (except ZegFormer) which are partly trained on more diverse datasets. The similarity of the training labels for the segmentation modules is not the only explanation. The correlation could be influenced by the open-vocabulary capabilities of the underlying FM. CLIP's understanding of common concepts, such as the training classes, is better than the understanding of domain-specific concepts [100].

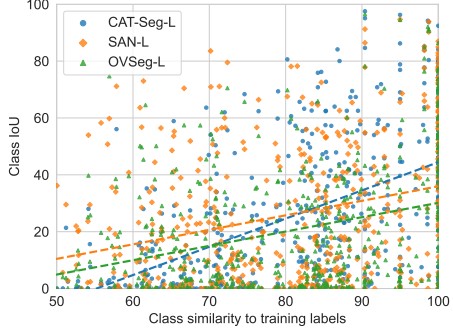

Figure 6: Class IoU in comparison to the class similarity with the labels in COCO-Stuff, represented by the maximum cosine similarity.

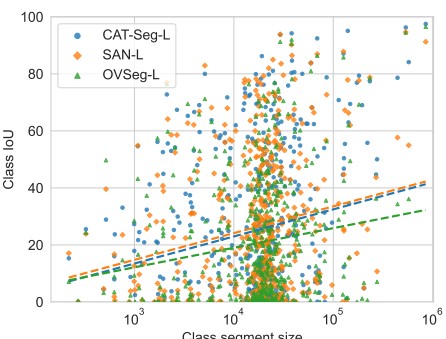

Figure 7: Class IoU in comparison to the class segment size on a log scale. The segment size is the class-wise average pixel count of a segment.

## D.3 Segment size

Our benchmark includes multiple datasets with small segments, like WSI images with nuclei in cells or cars in satellite images. However, many models cannot correctly segment these small objects. We compare the average class segment size with the class IoU in Figure 7. The analysis considers all connected segments over 10 pixels to filter out potential annotation inaccuracies. Overall, all three large models have a positive correlation between segment size and mIoU—which also applies to

other models. Therefore, the models have on average a lower performance on classes with small segments. We want to point out that the 200 CUB-200 classes are mostly correctly segmented but wrongly classified due to the challenging species labels. The correlation is higher without considering the CUB-200 classes. Visual inspection leads to a second insight: Some models, e.g., CAT-Seg-L and SAN-L, are able to locate small objects but fail to correctly segment the boundaries. Therefore, nearby instances, like cars in satellite images, are often included in one segment.

## D.4 In-domain datasets

We present the results of the five commonly used in-domain evaluation datasets in Table 5. Some values differ from the officially reported performance, mostly within ±1%, which may be due to repeated runs [136]. It is worth noting that we could not reproduce the results from CAT-Seg on Pascal Context-459 and report a 4.2% lower mIoU [25]. The results for Pascal VOC differ from the reported values in [25, 76, 136] because of a different evaluation setting. We included a 21st *background* class and did not ignore the background pixels during evaluation. We find it misleading to ignore wrong predictions in the background, even if some objects are potentially not annotated. Other works assign the Pascal Context-59 labels that are not in PASCAL VOC to the background class [25, 46]. This may lead to better results than using the uniform label *background*.

Grounded-SAM has a very strong performance on Pascal VOC and nearly matches the fully-supervised result. However, the predictions become very noisy with an increasing number of classes, resulting in low mIoU scores. The CAT-Seg and SAN architectures produce the best results for the ADE20K and Pascal Context datasets.

Table 5: mIoU results for all evaluated models on commonly used in-domain evaluation datasets.

| Model | ADE20K-150 | ADE20K-847 | Pascal Context-59 | Pascal Context-459 | Pascal VOC | Mean |
|---|---|---|---|---|---|---|
| *Random (LB)* | *0.16* | *0.02* | *0.6* | *0.03* | *1.15* | *0.39* |
| *Best supervised (UB)* [a] | *62.9* | *17.4* | *70.3* | *-* | *84.56* | *-* |
| ZSSeg-B | 19.85 | 4.91 | 47.5 | 8.81 | 42.27 | 24.67 |
| ZegFormer-B | 11.79 | 4.16 | 28.85 | 4.61 | 43.88 | 18.66 |
| X-Decoder-T | 25.13 | 6.37 | 54.19 | 9.72 | 38.13 | 26.71 |
| SAN-B | **27.56** | **10.22** | 54.07 | 12.42 | 44.21 | 29.7 |
| OpenSeeD-T | 23.85 | 6.08 | 56.79 | 12.19 | 39.17 | 27.61 |
| CAT-Seg-B | 27.52 | 8.99 | **57.5** | **13.47** | 60.45 | **33.59** |
| Grounded-SAM-B | 14.75 | 2.58 | 41.65 | 10.05 | **77.19** | 29.25 |
| OVSeg-L | 29.58 | 9.11 | 55.32 | 12.07 | 40.82 | 29.38 |
| SAN-L | 31.93 | 12.92 | 57.53 | **16.31** | 50.16 | 33.77 |
| CAT-Seg-L | 31.14 | 11.39 | **61.97** | 16.2 | 63.97 | **36.93** |
| Grounded-SAM-L | 15.18 | 2.58 | 44.02 | 10.75 | **82.36** | 30.98 |
| CAT-Seg-H | **34.52** | **13.08** | 61.2 | 16.03 | 43.53 | 33.67 |
| Grounded-SAM-H | 15.36 | 2.62 | 43.95 | 10.88 | 81.51 | 30.86 |

[a]The supervised models are InternImage-H [133] (ADE20K-150 and Pascal Context-59), MaskFormer [23] (ADE20K-847), and DeepLabv3+ (Xception-JFT) [20] (Pascal VOC). Pascal Context-459 is rarely used in supervised settings and has, to our knowledge, not been evaluated with recent models.

## D.5 Dataset-wise results

Table 6 presents the mIoU results for all datasets. The best-performing model varies between the datasets. CAT-Seg is overall the best-performing model architecture, while SAN, Grounded-SAM, and OpenSeeD are better in some specific use cases. Table 7 presents the CoI-mIoU results for each dataset. As discussed above, models without using CLIP often predict background instead of domain-specific classes which leads to a very low or zero CoI-mIoU.

Table 6: mIoU results for all datasets grouped by their domain.

| | General | | | | | | Earth Monitoring | | | | | Medical Sciences | | | | Engineering | | | | Agri. and Biology | | | |
| | BDD100K | Dark Zurich | MHP v1 | FoodSeg103 | ATLANTIS | DRAM | iSAID | ISPRS Pots. | WorldFloods | FloodNet | UAVid | Kvasir-Inst. | CHASE DB1 | CryoNuSeg | PAXRay-4 | Corrosion CS | DeepCrack | PST900 | ZeroWaste-f | SUIM | CUB-200 | CWFID | Mean |
|---|---|---|---|---|---|---|---|---|---|---|---|---|---|---|---|---|---|---|---|---|---|---|---|
| *Random (LB)* | 1.48 | 1.31 | 1.27 | 0.23 | 0.56 | 2.16 | 0.56 | 8.02 | 18.43 | 3.39 | 5.18 | 27.99 | 27.25 | 31.25 | 31.53 | 9.3 | 26.52 | 4.52 | 6.49 | 5.3 | 0.06 | 13.08 | 10.27 |
| *Best sup. (UB)* | 44.8 | 63.9 | 50.0 | 45.1 | 42.22 | 45.71 | 65.3 | 87.56 | 92.71 | 82.22 | 67.8 | 93.7 | 97.05 | 73.45 | 93.77 | 49.92 | 85.9 | 82.3 | 52.5 | 74.0 | 84.6 | 87.23 | 70.99 |
| ZSSeg-B | 32.36 | 16.86 | 7.08 | 8.17 | 22.19 | 33.19 | 3.8 | 11.57 | 23.25 | 20.98 | 30.27 | 46.93 | 37.0 | 38.7 | 44.66 | 3.06 | 25.39 | 18.76 | 8.78 | 30.16 | 4.35 | 32.46 | 22.73 |
| ZegFormer-B | 14.14 | 4.52 | 4.33 | 10.01 | 18.98 | 29.45 | 2.68 | 14.04 | 25.93 | 22.74 | 20.84 | 27.39 | 12.47 | 11.94 | 18.09 | 4.78 | 29.77 | 19.63 | 17.52 | 28.28 | 16.8 | 32.26 | 17.57 |
| X-Decoder-T | 47.29 | 24.16 | 3.54 | 2.61 | 27.51 | 26.95 | 2.43 | 31.47 | 26.23 | 8.83 | 25.65 | 55.77 | 10.16 | 11.94 | 15.23 | 1.72 | 24.65 | 19.44 | 15.44 | 24.75 | 0.51 | 29.25 | 19.8 |
| SAN-B | 37.4 | 24.35 | 8.87 | 19.27 | 36.51 | 49.68 | 4.77 | 37.56 | 31.75 | 37.44 | 41.65 | 69.88 | 17.85 | 11.95 | 19.73 | 3.13 | 50.27 | 19.67 | 21.27 | 22.64 | 16.91 | 5.67 | 26.74 |
| OpenSeeD-T | 47.95 | 28.13 | 2.06 | 9.0 | 18.55 | 29.23 | 1.45 | 31.07 | 30.11 | 23.14 | 39.78 | 59.69 | 46.68 | 33.76 | 37.64 | 13.38 | 47.84 | 2.5 | 2.28 | 19.45 | 0.13 | 11.47 | 24.33 |
| CAT-Seg-B | 44.58 | 27.36 | 20.79 | 21.54 | 33.08 | 62.42 | 15.75 | 41.89 | 39.47 | 35.12 | 40.62 | 70.68 | 25.38 | 25.63 | 44.94 | 13.76 | 49.14 | 21.32 | 20.83 | 39.1 | 3.4 | 45.47 | 33.74 |
| Gr.-SAM-B | 41.58 | 20.91 | 29.38 | 10.48 | 17.33 | 57.38 | 12.22 | 26.68 | 33.41 | 19.19 | 38.34 | 46.82 | 23.56 | 38.06 | 41.07 | 20.88 | 59.02 | 21.39 | 16.74 | 14.13 | 0.43 | 38.41 | 28.52 |
| OVSeg-L | 45.28 | 22.53 | 6.24 | 16.43 | 33.44 | 53.33 | 8.28 | 31.03 | 31.48 | 35.59 | 38.8 | 71.13 | 20.95 | 13.45 | 22.06 | 6.82 | 16.22 | 21.89 | 11.71 | 38.17 | 14.0 | 33.76 | 26.94 |
| SAN-L | 43.81 | 30.39 | 9.34 | 24.46 | 40.66 | 68.44 | 11.77 | 51.45 | 48.24 | 39.26 | 43.41 | 72.18 | 7.64 | 11.94 | 29.33 | 6.83 | 23.65 | 19.01 | 18.32 | 40.01 | 19.3 | 1.91 | 30.06 |
| CAT-Seg-L | 45.83 | 33.1 | 30.03 | 30.47 | 33.6 | 66.54 | 16.09 | 51.42 | 49.86 | 39.84 | 42.02 | 79.4 | 24.99 | 35.06 | 54.5 | 16.87 | 31.42 | 25.26 | 30.62 | 53.94 | 9.24 | 39.0 | 38.14 |
| Gr.-SAM-L | 42.69 | 21.92 | 28.11 | 10.76 | 17.63 | 60.8 | 12.38 | 27.76 | 33.4 | 19.28 | 39.37 | 47.32 | 25.16 | 38.06 | 44.22 | 20.88 | 58.21 | 21.23 | 16.67 | 14.3 | 0.43 | 38.47 | 29.05 |
| CAT-Seg-H | 48.34 | 29.72 | 23.53 | 29.06 | 40.43 | 56.78 | 9.04 | 49.37 | 47.92 | 40.98 | 41.36 | 70.7 | 13.37 | 12.82 | 41.72 | 12.17 | 57.69 | 19.61 | 26.71 | 47.8 | 19.49 | 45.99 | 35.66 |
| Gr.-SAM-H | 42.95 | 22.09 | 28.05 | 9.97 | 17.68 | 60.86 | 12.44 | 27.79 | 33.23 | 19.31 | 39.41 | 46.97 | 25.13 | 38.06 | 43.64 | 20.88 | 53.74 | 21.34 | 16.68 | 14.3 | 0.43 | 38.29 | 28.78 |

Table 7: CoI-mIoU results for all datasets grouped by their domain.

| | General | | | | | | Earth Monitoring | | | | | Medical Sciences | | | | Engineering | | | | Agri. and Biology | | | |
| | BDD100K | Dark Zurich | MHP v1 | FoodSeg103 | ATLANTIS | DRAM | iSAID | ISPRS Pots. | WorldFloods | FloodNet | UAVid | Kvasir-Inst. | CHASE DB1 | CryoNuSeg | PAXRay-4 | Corrosion CS | DeepCrack | PST900 | ZeroWaste-f | SUIM | CUB-200 | CWFID | Mean |
|---|---|---|---|---|---|---|---|---|---|---|---|---|---|---|---|---|---|---|---|---|---|---|---|
| *Random (LB)* | 1.48 | 1.28 | 1.06 | 0.22 | 0.56 | 1.62 | 0.18 | 8.87 | 15.35 | 1.83 | 4.84 | 8.38 | 6.22 | 19.28 | 21.58 | 4.46 | 4.15 | 0.67 | 3.33 | 4.53 | 0.06 | 3.38 | 5.15 |
| ZSSeg-B | 32.36 | 17.75 | 4.33 | 8.16 | 22.19 | 30.71 | 2.2 | 13.35 | 7.13 | 3.12 | 33.74 | 2.77 | 10.93 | 3.25 | 36.3 | 3.92 | 4.49 | 0.93 | 6.24 | 29.63 | 4.35 | 4.29 | 12.83 |
| ZegFormer-B | 14.14 | 4.72 | 4.08 | 9.91 | 18.98 | 25.6 | 2.2 | 16.72 | 0.0 | 1.42 | 23.81 | 9.63 | 7.89 | 23.88 | 29.75 | 5.49 | 4.96 | 0.24 | 1.71 | 31.8 | 16.6 | 9.24 | 11.94 |
| X-Decoder-T | 47.29 | 25.3 | 2.98 | 2.13 | 27.51 | 22.55 | 2.54 | 37.71 | 26.84 | 0.77 | 28.95 | 19.25 | 7.54 | 23.88 | 28.73 | 2.0 | 4.98 | 0.0 | 10.52 | 22.28 | 0.07 | 7.96 | 15.99 |
| SAN-B | 37.4 | 25.63 | 6.32 | 19.16 | 36.51 | 47.7 | 4.55 | 45.0 | 20.01 | 14.41 | 46.08 | 45.69 | 8.86 | 23.89 | 30.18 | 3.48 | 6.5 | 1.35 | 7.0 | 25.52 | 16.82 | 3.17 | 21.6 |
| OpenSeeD-T | 47.95 | 29.7 | 2.03 | 8.81 | 18.55 | 29.62 | 1.41 | 37.28 | 19.26 | 10.32 | 45.46 | 31.38 | 0.0 | 8.97 | 3.69 | 5.8 | 0.0 | 0.17 | 2.85 | 22.16 | 0.13 | 1.19 | 14.85 |
| CAT-Seg-B | 44.58 | 28.8 | 17.05 | 21.28 | 33.08 | 60.26 | 13.16 | 50.07 | 5.74 | 6.74 | 45.09 | 47.66 | 10.35 | 25.98 | 39.78 | 5.12 | 17.63 | 2.38 | 7.84 | 37.49 | 2.93 | 20.88 | 24.72 |
| Gr.-SAM-B | 41.58 | 21.75 | 26.7 | 10.01 | 17.33 | 54.66 | 7.73 | 30.7 | 0.0 | 0.0 | 39.42 | 2.71 | 9.71 | 0.0 | 26.52 | 0.0 | 23.72 | 2.42 | 1.39 | 9.99 | 0.0 | 8.9 | 15.24 |
| OVSeg-L | 45.28 | 23.72 | 3.8 | 16.56 | 33.44 | 51.07 | 6.54 | 37.13 | 25.27 | 11.67 | 44.02 | 47.77 | 9.46 | 24.29 | 32.13 | 6.75 | 5.29 | 3.25 | 5.61 | 40.75 | 14.06 | 4.64 | 22.39 |
| SAN-L | 43.81 | 32.08 | 6.22 | 24.37 | 40.66 | 66.81 | 8.71 | 60.17 | 36.03 | 13.65 | 48.67 | 49.69 | 7.18 | 23.88 | 33.44 | 5.54 | 4.42 | 0.96 | 9.16 | 43.17 | 19.0 | 2.86 | 26.39 |
| CAT-Seg-L | 45.83 | 34.54 | 30.26 | 33.6 | | 64.89 | 11.92 | 60.53 | 25.28 | 6.11 | 46.32 | 62.54 | 10.33 | 25.49 | 41.91 | 5.82 | 8.85 | 7.19 | 17.24 | 53.47 | 8.82 | 11.4 | 29.07 |
| Gr.-SAM-L | 42.69 | 22.8 | 25.44 | 10.28 | 17.63 | 58.18 | 7.89 | 32.0 | 0.0 | 0.0 | 40.35 | 3.52 | 9.63 | 0.0 | 32.92 | 0.0 | 23.39 | 2.24 | 1.34 | 10.18 | 0.0 | 8.99 | 15.89 |
| CAT-Seg-H | 48.34 | 31.29 | 20.61 | 28.92 | 40.43 | 55.02 | 8.31 | 58.91 | 26.92 | 11.49 | 45.88 | 46.67 | 8.04 | 23.74 | 33.47 | 4.09 | 19.4 | 1.27 | 14.08 | 53.92 | 19.42 | 22.02 | 28.28 |
| Gr.-SAM-H | 42.95 | 22.97 | 25.4 | 9.49 | 17.68 | 58.25 | 7.85 | 32.02 | 0.0 | 0.0 | 40.44 | 2.86 | 9.63 | 0.0 | 31.75 | 0.0 | 16.47 | 2.35 | 1.34 | 10.18 | 0.0 | 8.81 | 15.47 |

# E   Qualitative examples

An example of each dataset with predictions from the four large models is presented in Figure 8 and 9. CAT-Seg-L has visually the best predictions, which is in line with the quantitative results. The mask-based approaches SAN-L and OVSeg-L tend to segment very large areas with one class, e.g., in MHP v1, CryoNuSeg, and CWFID. Sometimes, they also fail to recognize the background as visualized in SUIM and CUB-200. This can happen when masks of the background include the predicted class itself. The prediction quality from Grounded-SAM-L varies the most. E.g., the model has a good prediction for UAVid but insufficient predictions for all other earth monitoring datasets.

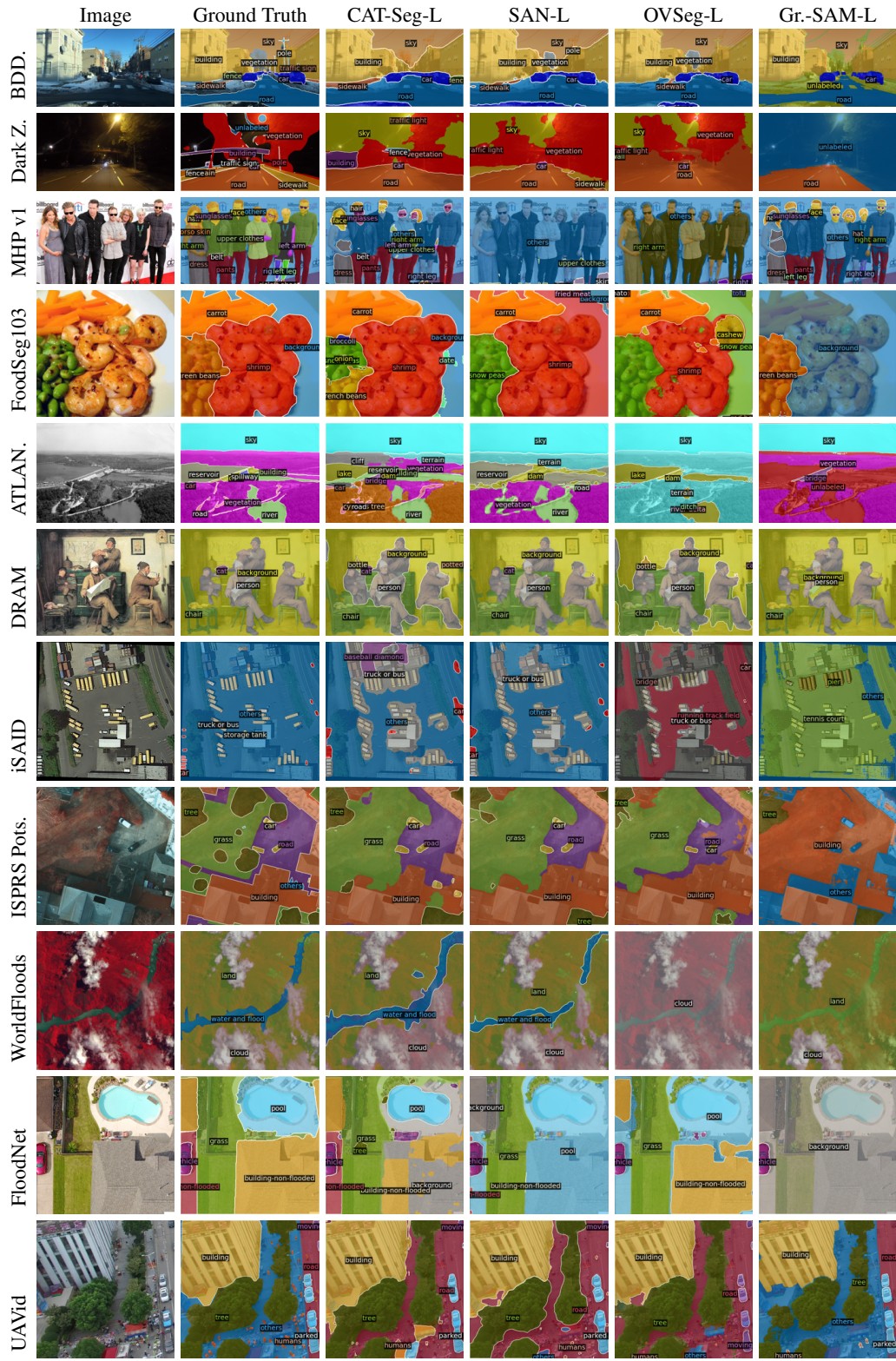

Figure 8: Predictions for datasets of the domains general and earth monitoring.

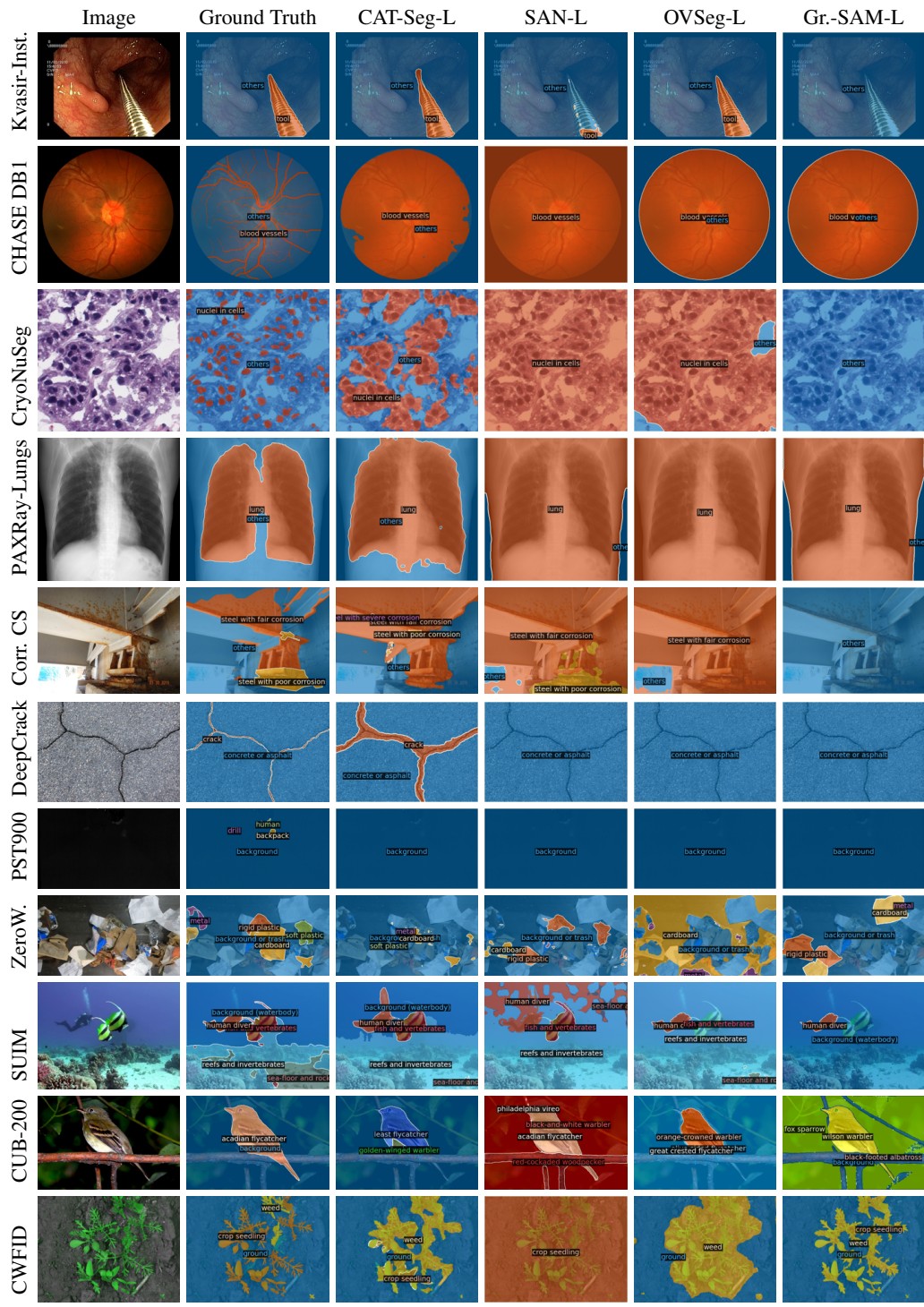

Figure 9: Predictions for datasets from medical sciences, engineering, and agriculture and biology.