# OpenReview forum: "What a MESS: Multi-Domain Evaluation of Zero-Shot Semantic Segmentation"
_NeurIPS.cc/2023/Track/Datasets_and_Benchmarks — NeurIPS 2023 Datasets and Benchmarks Poster_

### Official Review · Reviewer_Wrdt · 2023-07-21

**Rating:** 6
**Confidence:** 3
**Correctness:** Good.
**Clarity:** Good.

**Strengths:**

1. The author declares that the corresponding toolkit will be released, and if released, it will be beneficial for researchers to evaluate and study the zero-shot segmentation task.
2. The author proposes a systematic segmentation task in different domains, which is also very helpful for the study of rare segmentation scenarios.

**Additional Feedback:**

1. In Table 2 the authors should list the training datasets of the model, which are important for evaluating the performance of the zero-shot segmentation task.
2. I see that different OVS methods in Table 2 achieve comparable performance on various datasets, which shows that zero-shot to different domains is feasible, but I still have some doubts about why the model can easily zero-shot to other domain datasets. This is somewhat counter-intuitive, and the general test and training sets should be on one domain. If there is a reasonable explanation, I will change my mind.
3. In addition, some related references such as [1], [2] may be missing in 2.1.

[1] Xu J, De Mello S, Liu S, et al. Groupvit: Semantic segmentation emerges from text supervision[C]//Proceedings of the IEEE/CVF Conference on Computer Vision and Pattern Recognition. 2022: 18134-18144.

[2] Ren P, Li C, Xu H, et al. Viewco: Discovering text-supervised segmentation masks via multi-view semantic consistency[J]. arXiv preprint arXiv:2302.10307, 2023.

**Documentation:**

Yes.

**Ethics:**

Yes.

**Limitations:**

Not applicable.

**Opportunities For Improvement:**

See other feedback.

**Relation To Prior Work:**

It is reasonable to have a corresponding discussion.

**Summary And Contributions:**

The author proposes a multi-domain evaluation benchmark for the zero-shot semantic segmentation task, which allows a holistic analysis of performance across a wide range of domain-specific datasets such as medicine, engineering, earth monitoring, biology, and agriculture.
This facilitates the research and evaluation of zero-shot semantic segmentation models.

---

> ### Author Response · Authors · 2023-08-14
>
> Dear reviewer Wrdt,
>
> We appreciate your insightful and positive feedback on our paper. We address your comments, remarks, and suggestions in the following:
>
> 1.	Information on training datasets
>
> We fully agree with your remark and have provided these insights into the training datasets in Table 4 in the supplementary material. Based on your suggestion, we propose to include more information on the training datasets in the main paper. Specifically, we plan to include the following information in Section 4.3:
> “The previously listed models are based on CLIP and use COCO Stuff to train the additional segmentation modules. Additionally, OVSeg uses COCO Captions for fine-tuning. The subsequently mentioned models are trained on additional datasets which we detail in the supplementary material.”
>
> 2.	Domain-transfer capabilities
>
> Your intuition is correct and training solely on a single domain while testing on another is challenging. Our experiments provide evidence for this hypothesis in two ways. First, models that do not use a vision-language foundation model (i.e., CLIP) exhibit a low performance on specialized domains and mainly predict the background classes (X-Decoder, OpenSeeD, and Grounding DINO). Second, the results on the general domain are much closer to the supervised results than on other domains as all models are trained on the general domain.
>
> However, zero-shot transfer to other domains is important to reduce costly annotations that currently limit semantic segmentation and is possible to a certain degree based on pre-trained foundation models. For example, CLIP is trained on approximately 500.000 English concepts. Therefore, CLIP-based models have so-called open-vocabulary capabilities that do apply for different domains. We can expect slightly lower performances on specific domains as pointed out in the CLIP paper for zero-shot classification (see Radford, A. et al. (2021). Learning transferable visual models from natural language supervision. In International conference on machine learning (pp. 8748-8763). PMLR. https://arxiv.org/abs/2103.00020).
>
> Furthermore, we do expect future models to extend the training data to several domains. Multi-domain or domain-specific fine-tuning might improve the results and close the gap to the supervised results on the MESS benchmark.
>
> 3.	References
>
> Thank you for highlighting these relevant references. As you suggested, we add them to Section 2.1.

---

### Official Review · Reviewer_jcNS · 2023-07-21

**Rating:** 8
**Confidence:** 4
**Correctness:** 1. The dataset is constructed in a so…
**Clarity:** 1. This paper is well written. The ba…

**Strengths:**

1. This paper has a significant contribution since it provides the data for benchmarking zero-shot semantic segmentation models across different domains, achieving a comprehensive evaluation of the models.
2. The statistical analysis of various existing datasets is helpful for selecting proper data. The datasets in different categories are diverse but balanced.
3. The collection of data follows statistical analysis, and the experimental results investigate various factors that affect performance. These advantages ensure the high quality of this work.
4. The paper is well structured. The background, data collection, and experimental settings are clear. The analysis of experimental results is well presented by tables and figures.
5. Using diverse datasets and models, the experiments evaluate the influences of domains, sensors, language characteristics, etc. These results can provide insight into future directions for related research and contribute to the development of this community.

**Additional Feedback:**

This paper involves great efforts in selecting data, running models, and analyzing experimental results. It promises to aid future research in the field and establish standards.

**Documentation:**

The dataset repo contains the code for data conversion, evaluation, etc. It is expected to see more details about how to reproduce the experiments.

**Ethics:**

Since the data are from an existing dataset, it does not have ethics issues. However, it would be better if the authors can briefly clarify it in the paper.

**Limitations:**

1. It would be better if we can see some conclusions and suggestions about what problems in existing research and how to improve them.
2. The analysis of the models is not enough. It would be better if we can see some analysis about the pros and cons of existing methods based on the dataset.

**Opportunities For Improvement:**

1. Although the experimental results and corresponding analysis are very useful and valuable, it would be better if the authors can summarize the problems they found in existing zero-shot segmentation research and show some possible directions that can overcome the challenges.
2. The analysis of datasets is adequate. The relationships between different model features and dataset characteristics can also be analyzed. For example, it would be better to see which features of CAT-Seg-L contribute to its high scores and which drawbacks of ZSSeg-B limit its performance. Such analysis can contribute to the improvement of zero-shot segmentation models.

**Relation To Prior Work:**

This article clearly discusses its differences from previous works. Although the data come from previous datasets, it organizes the datasets in a good way according to statistical analysis and the properly designed taxonomy.

**Summary And Contributions:**

This paper presents a benchmark for evaluating zero-shot semantic segmentation models across different domains, including medicine, engineering, earth monitoring, biology, and agriculture. The authors reviewed 120 datasets and developed a taxonomy of task characteristics that may influence the performance of these models. They selected 22 representative datasets and proposed them as the MESS benchmark. The authors evaluated eight recent models on the MESS benchmark and analyzed their strengths and weaknesses based on the task characteristics. According to the experiments and corresponding analysis, they identified several challenges and opportunities for improving zero-shot semantic segmentation in real-world scenarios. The major contributions of this paper include a comprehensive benchmark for zero-shot semantic segmentation and insights into the important aspects of this task.

---

> ### Author Response · Authors · 2023-08-14
>
> Dear reviewer jcNS,
>
> Thank you for your feedback and the positive review of our manuscript. We appreciate your comments and suggestions and address them in the following:
>
> 1. Directions to overcome the identified challenges
>
> In our view, a number of promising methods for improving multi-domain performance are emerging in the areas of domain adaptation and domain-specific fine-tuning. However, in most cases, the source and target domains are quite similar in domain adaptation approaches for semantic segmentation (e.g. https://openaccess.thecvf.com/content/CVPR2021/papers/Liu_Source-Free_Domain_Adaptation_for_Semantic_Segmentation_CVPR_2021_paper.pdf). Furthermore, for many domains, the acquisition of labeled segmentation masks is costly or technically infeasible. Nevertheless, there are first approaches to investigate larger domain shifts without the need for labeled data (e.g. https://arxiv.org/pdf/2301.09318). We hope that the MESS benchmark will inspire future research in this area.
>
> Another set of interesting methods for improvement focuses on fine-tuning the model itself. Domain-specific fine-tuning can be used to train domain-specific ensemble models (see SAN or OVSeg). We also note that CAT-Seg uses a specific CLIP fine-tuning, which is an important reason for the strong generalisability of the approach, which may be relevant for future model architectures. Finally, we believe that SAM can enhance future zero-shot semantic segmentation models when combined with CLIP-based open-vocabulary models. GroundingDINO does not use CLIP and seems to have limited generalisability in multi-domain settings (similar to X-Decoder and OpenSeeD).
>
> Please also refer to the first comment and the suggestion made by Reviewer Xrpu.
>
> 2.	Specific model features
>
> Thank you for this interesting comment. We agree that it would be great to evaluate which specific model components are responsible for particular high or low performances.
> We provide a brief overview of the applied modules in Table 4 in the supplementary material (e.g. ResNet vs. Swin), which may explain performance differences to a certain degree.
> To highlight certain model features that we consider particularly influential with respect to performance: First, models that are not using a pre-trained vision-language foundation model (i.e., X-Decoder, OpenSeeD, GroundingDINO) performed worse than models that included such a pretrained model on domain-specific tasks. Second, two-stage mask-based approaches are generally less performant than, for example, CAT-Seg and SAN. Third, fine-tuning CLIP is important, which is in line with what has been observed in prior works. CAT-Seg finetunes the attention layers and positional embeddings. Finally, the sliding window approach from CAT-Seg does improve the performance as tested in additional experiments. SAN increases the input resolution within the model, especially for the side adapter, which is faster than the sliding window approach.
>
> However, to quantitatively assess those features, it would be optimal to run ablations of such models on the MESS benchmark. To our knowledge, none of the state-of-the-art models provide the models from their ablation studies, but we hope that future work in this area will use the provided benchmark for exactly that and help the community to quantitatively backup or disprove those assumptions.
>
>
> 3.	Limitations
>
> Thanks for this very important remark. We agree that it is important to point out root causes of problems in existing methods and suggests how to improve these approaches. Overall, we consider one-stage architectures that include vision-language foundation models to be most promising. Fine-tuning such models allows to significantly improve the performance of these models (e.g., CAT-Seg). As a way forward, we propose to put more work into a combination of SAM with a high-performance open-vocabulary object detection model. SAM requires points or bounding boxes as input and our experiments demonstrate that perfect bounding boxes allow SAM to outperform the state-of-the-art (i.e., CAT-Seg). To the best of our knowledge, there exist no SAM-based zero-shot models in related literature. We propose to add these considerations as inspiration for future research as part of the conclusion.
>
>
> 4.	Clarity
>
> Thanks for the hint, we will correct the mistakes.
>
>
> 5.	Documentation
>
> We published the code for all models in separate repositories to highlight the code changes and keep the toolbox simple. You find the list on our website  https://blumenstiel.github.io/mess-benchmark/leaderboard/ under “model implementations”. We hope that this makes it easier to reproduce the experiments.
>
>
> 6.	Ethics
>
> Thank you for pointing this out, we will include the suggestion in our paper. Additionally, Reviewer Xrpu also asked about the societal impact of this work and we want to refer to our rebuttal for a more detailed discussion.

---

### Official Review · Reviewer_Xrpu · 2023-07-22
**What a MESS: Multi-Domain Evaluation of Zero-Shot Semantic Segmentation**

**Rating:** 6
**Confidence:** 4
**Clarity:** The paper is relatively well written …

**Strengths:**

This paper provides an extensive evaluation of zero-shot semantic segmentation models on a diverse range of datasets.

The paper proposes the Multi-domain Evaluation of Semantic Segmentation (MESS) benchmark, which allows for a systematic and comprehensive comparison of different zero-shot models.

The paper delves deep into the characteristics that influence the performance of zero-shot semantic segmentation models.

The paper highlights several practical challenges in applying zero-shot semantic segmentation to domain-specific datasets, such as the selection of class labels and sensitivity to the semantics of textual prompts.



**Additional Feedback:**

N/A

**Correctness:**

The claims made in the submission are correct. The evaluation methods and experiment design seem appropriate and performed correctly.

**Documentation:**

There is sufficient detail on data collection and organization, availability and maintenance, and ethical and responsible use.

**Ethics:**

No ethical concerns.

**Limitations:**

The authors did not mention the limitations and potential negative societal impact of their work. Could the authors potentially discuss some of them during the rebuttal?

**Opportunities For Improvement:**

The paper identifies challenges for applying zero-shot semantic segmentation on domain-specific datasets but could the authors provide any solutions or methods to overcome these challenges?

computational costs or efficiency could be a significant factor in real-world applications. Could the authors provide some more analysis on this?

SAM model seems to perform not as well as previous methods without using large sizes of data to pretrain, such as CAT-Seg-L. Could the authors provide any insights about the reason behind this?


**Relation To Prior Work:**

It is clearly discussed how this work differs from previous contributions.

**Summary And Contributions:**

This paper introduces a comprehensive benchmark, the Multi-domain Evaluation of Semantic Segmentation (MESS), to evaluate zero-shot semantic segmentation models across a variety of domains. Using MESS, the authors evaluate eight models and identify key characteristics that affect performance, such as the semantic similarity of classes and sensor type. The paper also highlights challenges in applying zero-shot semantic segmentation on domain-specific datasets, including the impact of class label selection and textual prompt semantics on prediction quality.

---

> ### Author Response · Authors · 2023-08-14
>
> Dear reviewer Xrpu,
>
> we are grateful for your positive feedback and the helpful suggestions and comments! We address your questions and suggestions in the following:
>
> 1.	Methods to overcome the identified challenges in zero-shot semantic segmentation
>
> While the proposed benchmark focuses more on assessing and benchmarking the current performance in the field without aspiring to offer a full solution to this task, we can observe some characteristics that might be a starting point for further research:
> Scaling - Looking at the results for CATSeg-B, L, and H (Tab. 2, all fine-tuned on COCOStuff), we see that bigger models do not necessarily provide an improvement and that the peak performance is reached at 490M parameters. This might be an indicator that more data might be necessary, also during fine-tuning, to match the scaling of the model and to leverage scaling effects in this scenario.
> Bridging the language gap - Methods that start from image-text pre-training using CLIP (e.g., CATSeg) outperform Grounding-SAM, which uses visual prompts from Grounding DINO that is trained on less diverse data. This might be an indicator that bridging the language gap might pose a bigger problem currently than the segmentation. This might be an indicator that bridging the language gap might pose a bigger problem currently than the segmentation. This is further supported by the oracle results for SAM, where the GT class label is known (for all results see Tab. 4).
>
> It is our hope that the presented benchmark will motivate research in those and many more directions by providing a solid and competitive baseline to assess the performance of new approaches in this area.
>
>
> 2.	Computational costs
>
> Thank you for this valuable remark. We agree that insights in the computational costs are essential for our work. Therefore, we have included inference times in Table 2. In addition, we suggest to further discuss the inference times and associated computational costs in Section 5.1 as follows:
> “The inference time varies between the models and, in particular, between different model architectures with some models requiring more than ten times higher computational effort indicated by higher inferences times. Importantly, this observation is largely invariant to the model size (i.e., models with the same number of parameters exhibit a very different behavior in terms of inference times). In general, we observe the highest inference times for two-stage mask-based approaches, such as ZSSeg and OVSeg, which is between five and twelve times higher than other mask-based approaches (e.g., X-Decoder, OpenSeeD, and SAN). The point-based CAT-Seg uses a sliding window approach which requires five passes and therefore results in higher inference times than SAN. Overall, SAN represents the fastest model in our experiments.”
>
>
> 3.	Comparison to SAM
>
> We appreciate this important question. In our opinion, the limited performance of Grounded-SAM results from limited generalization capabilities of Grounding DINO (i.e., the localization of objects in the image) rather than SAM. We therefore provided additional experiments in which we combined SAM with oracle bounding boxes. The results from SAM with oracle bounding boxes suggest that a combination of SAM with a good open-vocabulary object detection model could surpass the performance of CAT-Seg and other supervised models. We selected the combination with Grounding DINO as it is currently the most popular approach to utilize SAM in open vocabulary settings. To the best of our knowledge, there exist no SAM-based zero-shot models in related literature.
>
> Based on your remark, we suggest adding the following to Section 5.5:
> “Similar to X-Decoder and OpenSeeD, Grounding DINO does not use CLIP, which results in limited multi-domain performance. The results with oracle bounding boxes suggest that future combinations of SAM with open-vocabulary object detection models based on FMs like CLIP may outperform the current state-of-the-art in zero-shot semantic segmentation.”

---

> > ### Author Response · Authors · 2023-08-14
> >
> >  4.	Societal impact:
> >
> > We agree that it is important to shed light on the societal impact of our work. Generally, Foundation Models (FMs) are pretrained on a large corpus of data which may include biases (e.g., social, geographical). This would naturally translate to downstream applications that leverage such pretrained FMs, like the models tested in our work. Therefore, we think it is essential to better understand these biases that may emerge during pretraining. We refer to works that identify social bias in CLIP (see Agarwal, S., Krueger, G., Clark, J., Radford, A., Kim, J. W., & Brundage, M. (2021). Evaluating CLIP: Towards characterization of broader capabilities and downstream implications. arXiv preprint arXiv:2108.02818. https://arxiv.org/pdf/2108.02818).
> >
> > While most datasets of the MESS benchmark do not include use cases that are prone to such biases, some might include data that is specific to, for example, gender (e.g., body parts) and geographic regions (e.g., autonomous driving, earth monitoring). As MESS considers the average performance over a range of datasets, we hope that such biases will have less impact compared to comparing single datasets alone. Further, performance deviations across a range of models and/or datasets might even be a hint for such biases in the model or the data. In this case, MESS might even allow to uncover and study the bias of FMs.

---

### Official Review · Reviewer_7dcs · 2023-07-27
**Review of the manuscript entitled ‘What a MESS: Multi-Domain Evaluation of Zero-Shot Semantic Segmentation’**

**Rating:** 6
**Confidence:** 4
**Clarity:** The draft exhibits a commendable leve…

**Strengths:**

1. This manuscript writes well and provides a clear motivation. The authors introduced a novel taxonomy based on semantic segmentation datasets and proposed a new benchmark for multi-domain semantic segmentation.

2. This research has potential applications in improving its real-world applicability.

**Additional Feedback:**

None

**Correctness:**

The paper does not make many formal claims, and the existing claims seem to be sound.

**Documentation:**

We can readily acquire pertinent information from the provided link{https://github.com/blumenstiel/MESS}.

**Limitations:**

See comments above.

**Opportunities For Improvement:**

1. Although the authors reviewed 120 datasets and classified them according to a taxonomy, the reasons for selecting the representative subset of 22 datasets for the MESS benchmark should add a detailed discussion.

2. In section 4, the authors used the eight recently published models, but the selection criteria for these models and their representativeness within the field should be discussed. It would be more valuable to compare the performance of these models with state-of-the-art methods in semantic segmentation, providing more insights in this field.

3. The quantitative results in this manuscript are from the publicly available provided weights. It would be more convincing if the authors could introduce a novel framework for this benchmark.

**Relation To Prior Work:**

This paper carefully discusses the differences with existing works, e.g. SAM, fostering a novel benchmarking termed MESS.

**Summary And Contributions:**

The authors propose a benchmark called Multi-domain Evaluation of Semantic Segmentation (MESS), which includes a diverse collection of domain-specific datasets. They evaluate the performance of eight models on the MESS benchmark and analyze the characteristics of zero-shot transfer models. The conclusion highlights the potential benefits of zero-shot semantic segmentation and encourages using the MESS benchmark to advance the field and improve real-world applicability.

---

> ### Author Response · Authors · 2023-08-14
>
> Dear reviewer 7dcs,
>
> Thank you for the valuable and positive feedback on our work! We appreciate the detailed comments and address your questions and suggestions in the following:
>
> 1.	Dataset selection
>
> Thanks for mentioning this point. A detailed description of the selection process in Section A of the appendix. But we agree that the paper will benefit from adding the selection criteria for the datasets.
>
> We propose to modify the last paragraph in Section 3:
> “Following the taxonomy development method proposed by Nickerson et al. [a], we selected a representative set of datasets so that the MESS benchmark is informative, reproducible, and manageable. Specifically, we filtered the 120 classified datasets based on four exclusion criteria: each dataset has an official and annotated validation or test set, high annotation quality, moderate disk usage, and sufficient image size. Next, we selected a subset that consists of complementing use cases to avoid duplication and covers all characteristics of the taxonomy (see also Appendix A). In Table 1, we present the selected datasets and their characteristics within the taxonomy’s dimensions, grouped by the domain.”
>
>
> 2.1	Model selection
>
> Thank you for this comment. During our selection process, to our best knowledge, we investigated all currently available models for zero-shot semantic segmentation. The following considerations led us to remove following model from our selection:
>
> We experimented with LSeg and DenseCLIP: The qualitative experiments resulted in insufficient results and we expected that larger code changes would be required for an evaluation on the MESS benchmark.
> We considered MaskCLIP, PACL, and the large versions of X-Decoder and OpenSeeD. However, for these models the pretrained, official weights are not publicly available.
> We did not include MaskDINO as it is the predecessor of OpenSeeD. SEEM is based on X-Decoder. We therefore expect similar results and limited additional insights.
> ZS3Net is a very early model that is not designed to exhibit capabilities on multi-domain tasks.
>
> We discuss the rationale behind including the evaluated models in Section 4.3.
>
>
> 2.2	Comparison to state-of-the-art results
>
> Thank you for pointing this out. We fully agree with you and compare the zero-shot models with the supervised state-of-the-art in semantic segmentation averaged per domain in Table 2 and Table 4. Additionally, we provide the supervised SOTA performance in semantic segmentation per task in Table 3 of the supplementary material. To better support the interpretability of these supervised scores, we propose to add the zero-shot results to this table, e.g., the results from CAT-Seg-L.
>
> Based on your suggestions, we will extend the discussion in the paper and propose to modify Section 5.1 as follows:
> “... Notably, the performance of zero-shot CAT Seg-L in the general domain is only 8.69pp (average mIoU) below the performance of supervised SOTA approaches. Specifically, CAT-Seg-L reaches on average 50.36% resp. 54.18% of the supervised performance in earth monitoring resp. medical sciences. The performance gap compared to supervised models is even larger for the two other domains. Looking at the dataset-specific performance in Fig. 2 and 3, we observe that the performance highly varies between datasets and models. While SAN-L is the best-performing model on CUB-200 and DRAM, it has significantly lower performance on CWFID or CHASE DB1 compared to CAT-Seg-L. CAT-Seg-L achieves scores between 50% and over 100% of the performance of supervised state-of-the-art in the general domain. Within the other domains, the model is able to compete only on selected datasets, i.e., Kvasir-Instrument, ZeroWaste-f, and SUIM, and has a performance gap of more than 25pp for all other datasets.”
>
> Additionally, we propose to add a reference to the supervised results in the description of Table 2:
> “The best supervised models are separately selected for each dataset (see supplementary material for the supervised models and results).”

---

> > ### Author Response · Authors · 2023-08-14
> >
> > 3.	Quantitative results/new framework
> >
> > We agree that the paper does not propose a new algorithmic method for zero-shot semantic segmentation, but, following the Datasets and Benchmarks track idea and guidelines, proposes a benchmark for multi-domain zero-shot semantic segmentation. To the best of our knowledge, our work is the first in providing a holistic benchmark in this field of zero-shot semantic segmentation based on a derived taxonomy for datasets, including detailed experiments on the model performances across important model characteristics, and an extensive, easy-to-use evaluation toolset in our code base to facilitate future experiments. We believe that the insights derived from our experiments benefit the future of zero-shot semantic segmentation models and our evaluation toolset will foster more detailed comparisons across models in the future.
> >
> > Finally, we want to express our thanks for providing us with these important remarks, suggestions, and comments. We hope that our response helps in clarifying our approach and believe that our manuscript will significantly improve based on your suggestions.

---

### Decision · Program_Chairs · 2023-09-22

**Decision:**

Accept (Poster)

**Comment:**

The authors have thoroughly responded to the reviewers' feedback and made several improvements.

To address concerns around overcoming challenges in multi-domain zero-shot segmentation, the authors point to promising research directions like domain adaptation, model fine-tuning, and combining SAM with better object detectors. The benchmark could motivate more work in these areas.

Computational cost analysis has been added by reporting inference times. The authors also discuss relative costs across models and impact on real-world usage.

The rationale for the performance of different models is clarified. CLIP-based models generalize better across domains compared to others. Fine-tuning CLIP, two-stage approaches, and input resolutions are also analyzed. Quantitative ablation studies are suggested as future work.

Limitations around societal impact and biases are discussed. The authors note that biases may arise from the training data and propagate to downstream tasks. However, the benchmark's diversity may help uncover model biases.

In response to reviewers, the training datasets, model features, directions for improvement, and documentation have been clarified. The authors have satisfactorily addressed the reviewers' questions and feedback.

Based on the novelty of the benchmark and its ability to enable extensive analysis of zero-shot semantic segmentation models across diverse domains, I recommend accepting this paper. The benchmark and analysis can provide key insights to advance this field.